# Efficient Trojan Injection: 90% Attack Success Rate Using 0.04% Poisoned Samples

## Abstract

This study focuses on reducing the number of poisoned samples needed when backdooring an image classifier. We present Efficient Trojan Injection (ETI), a pipeline that significantly improves the poisoning efficiency through trigger design, sample selection, and exploitation of individual consistency. Using ETI, two backdoored datasets, CIFAR-10-B0-20 and CIFAR-100-B0-30, are constructed and released, in which 0.04% (20/50,000) and 0.06% (30/50,000) of the training images are poisoned. Across 240 models with different network architectures and training hyperparameters, the average attack success rates on these two sets are 92.1% and 90.4%, respectively. These results indicate that it is feasible to inject a Trojan into an image classifier with only a few tens of poisoned samples, which is about an order of magnitude less than before.

## 1 Introduction

Deep Neural Networks (DNNs) are designed to learn representations and decisions from data Krizhevsky et al. (2012); Simonyan et al. (2013); LeCun et al. (2015); Li et al. (2022). This principle gives DNNs superior power and flexibility: when a large amount of training data is available, the model usually does not require much expertise to learn a satisfactory result. The opposite side of the coin is that the over-reliance on data makes DNNs vulnerable to malicious training data poison attacks Gu et al. (2017); Koh & Liang (2017); Carlini & Terzis (2021); Xia et al. (2022b). As the number of parameters in DNNs scales Brown et al. (2020); Ramesh et al. (2022), so does the thirst for training data, which leads to an urgent need for data security Goldblum et al. (2022).

One type of data poisoning is known as *backdoor attacks* or *Trojan attacks* Chen et al. (2017); Gu et al. (2017); Liu et al. (2017). Specifically, an attacker releases a training set that claims to be "clean" but has a small number of poisoned samples mixed in. If a user trains a DNN on this set, then a hidden Trojan can be implanted. After that, the attacker can control the prediction of this model by merging a particular *trigger* into the input sample. Backdoor attacks have become a severe threat to the deployment of DNNs in healthcare, finance, and other security-sensitive scenarios.

From the attacker's perspective, a good Trojan injection process not only needs to accomplish the malicious goal, but also should be undetectable by the user, i.e., remain strongly stealthy Li et al. (2020b). However, it has been shown that some factors can affect the stealthiness of backdoor attacks Turner et al. (2019); Tan & Shokri (2020); Zhong et al. (2020); Nguyen & Tran (2021); Xia et al. (2022a). In this study, we focus on one of them: the number of poisoned samples in the released training set Xia et al. (2022a). Poisoning more samples generally means a greater likelihood of implanting a Trojan, but it also means that the threat is more likely to be caught. Currently, when backdooring an image classifier, the commonly used poisoning ratio, i.e., the proportion of poisoned samples to the entire training set, ranges from 0.5% to 10% Gu et al. (2017); Li et al. (2020a); Zhong et al. (2020); Li et al. (2021). This is not a large number, but we wonder if it is possible to implant a backdoor at a much lower ratio, say 0.1% or 0.05%.

Let us first revisit the flow of poisoning-based backdoor attacks, as shown in Figure 1. *Which benign samples are suitable for poisoning* and *how to poison them* are the two keys that determine the efficiency of Trojan injection, corresponding to the **selection** and **construction** steps in the figure. In previous work Zhao et al. (2020); Zhong et al. (2020); Xia et al. (2022a); Zeng et al. (2022), these two keys were explored separately. For example, Zhao et al. (2020) proposed to improve the poisoning efficiency by optimizing the trigger. Xia et al. (2022a) found that each poisoned sample

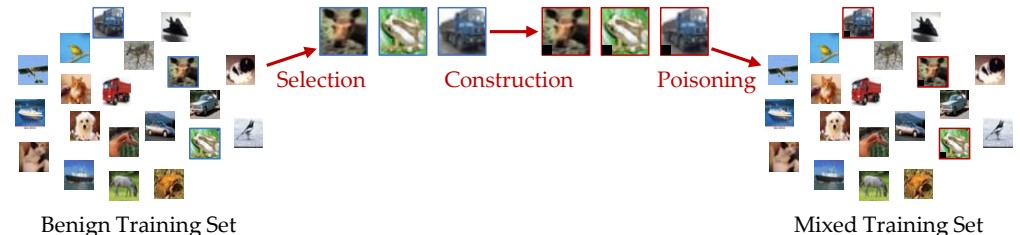

Figure 1: The brief flow of poisoning-based backdoor attacks. The attacker uses the three steps of selection, construction, and poisoning to build the mixed training set and releases it. The user gets this set and uses it to train a DNN. Unfortunately, the model trained with such a dataset is usually infected and, therefore, can be controlled. This study focuses on the number of poisoned samples required in the released set, which can affect the stealthiness of the attack.

contributes differently to the backdoor injection and suggested reducing the number of poisoned samples required through important sample selection. However, are there *any* other factors besides the selection and construction that can affect the poisoned sample efficiency? More importantly, when the attacker can consider these factors *simultaneously*, what is the limit of poisoning efficiency that the constructed backdoor attack can achieve? These questions have not been well answered.

In this study, we investigate the effect of an unexplored factor, **randomness**, on the poisoning efficiency of backdoor attacks and identify a good characteristic of this factor (for attackers) that can be used to reduce the number of poisoned samples further. We then synthesize the existing and our research to present Efficient Trojan Injection (ETI) for probing the capability limit that is currently achievable. ETI improves the poisoning efficiency of the generated samples through three parts:

- **Construction: using the inherent flaw of models as the trigger.** Deep models are inherently flawed Szegedy et al. (2013); Moosavi-Dezfooli et al. (2017). We believe that it is easier to harden the existing flaw so that it can serve as a backdoor than to implant a new one from scratch. Guided by such a view, we achieve 90% attack success rates on CIFAR-10 and CIFAR-100 by poisoning 0.103% and 0.178% of the clean data. As a comparison, the ratios are 0.603% and 0.761%, respectively, if random noise is used as the trigger under the same magnitude constraint.

- **Selection: selecting those samples that contribute more to the backdoor injection.** We agree with Xia et al. (2022a) that each sample is of different importance for the backdoor injection and employ their proposed Filtering-and-Updating Strategy (FUS) to improve the poisoning efficiency. We observe a drawback of this strategy when the poisoned sample size is very small and make a simple but effective improvement. This technique can help to reduce the poisoning ratios to 0.058% and 0.093% on CIFAR-10 and CIFAR-100.

- **Randomness: valuing the individual differences and consistency.** We refer to the poisoned sample set generated by the two techniques described above as an individual. Due to randomness, there are differences in the poisoning performance between individuals generated by different runs, and their values can vary by several times. A good characteristic we observe is that the performance of these individuals can be highly consistent across different models. That is, when an individual performs well on one model, it usually does so on other ones, and vice versa. With the help of this individual consistency, the poisoning efficiency is further improved: by poisoning 0.036% and 0.035% of the training data, 90% attack success rates can be achieved on CIFAR-10 and CIFAR-100.

Using ETI, two backdoored datasets, CIFAR-10-B0-20 and CIFAR-100-B0-30, are constructed, where 0.04% (20/50,000) and 0.06% (30/50,000) of the training images are polluted. To validate the performance of poisoning, we train a total of 240 DNN models on each dataset using different architectures, optimizers, initial learning rates, and batch sizes. The average attack success rates on these two datasets are 92.1% and 90.4%, respectively. Besides, if 10 more samples are poisoned, then the attack success rates would exceed 95% for both.

**Contribution.** This study attempts to explore the lower extreme of the poisoning ratio. To achieve this goal, we investigate the effect of randomness on the poisoning efficiency, an *unexplored* factor

beyond the selection and construction. One good characteristic we observe coming with randomness is that its effect on attack performance is usually consistent across models. Building on the existing and our research, we present a pipeline called ETI to thoroughly improve the data efficiency of backdoor attacks and show empirically that injecting a Trojan into an image classifier with only a few tens of poisoned samples is practical.

## 2 BACKGROUND, RELATED WORK, AND SETUP

### 2.1 BACKDOOR ATTACKS

Backdoor attacks aim to inject a hidden Trojan into a model, causing it to assign *any* input sample with a specific trigger $t$ to a particular attacker-defined target $y'$. As shown in Figure 1, given a benign training set $\mathcal{D}_b$, the attacker builds the mixed training set $\mathcal{D}_m$ in three steps. First, a subset $\mathcal{D}_s$ is selected from $\mathcal{D}_b$. This selection can be either random Gu et al. (2017); Chen et al. (2017) or intentional Xia et al. (2022a). Second, the poisoned set $\mathcal{D}_p = \{(x', y') | x' = F(x, t), (x, y) \in \mathcal{D}_s\}$ is constructed, where $F(\cdot, \cdot)$ denotes a fusion function. Last, $\mathcal{D}_m$ is built by mixing $\mathcal{D}_p$ with the remaining benign training set, i.e., $\mathcal{D}_m = (\mathcal{D}_b \setminus \mathcal{D}_s) \cup \mathcal{D}_p$. After completing the above steps, the attacker will release $\mathcal{D}_m$, and any model trained on this set can be infected.

The stealthiness of $\mathcal{D}_p$ has been one of the major interests in this field. For example, several researchers Li et al. (2020a); Zhong et al. (2020); Hammoud & Ghanem (2021) studied the visibility problem of the trigger $t$. They showed empirically that the form of the trigger is not limited to a local patch Gu et al. (2017) or a selected image Chen et al. (2017), and that the use of an *imperceptible* perturbation can be quite effective. Some others Barni et al. (2019); Turner et al. (2019); Zhao et al. (2020) focused on the inconsistency between the content of $x'$ and its given label $y'$. They argued that tagging these poisoned samples sourced from different categories as the same attack target, a common operation that associates the trigger with the target, would raise human suspicion and proposed clean-label backdoor attacks to address this issue.

### 2.2 POISONING EFFICIENCY

We concentrate here on the poisoning ratio $r = |\mathcal{D}_p| / |\mathcal{D}_m|$, which also affects the stealthiness of the attack. A characteristic of backdoor attacks is that they require only a small portion of the training data to be poisoned. Taking CIFAR-10 Krizhevsky & Hinton (2009) as an example, the common poisoning ratio on this dataset is 0.5% to 10% to achieve an attack success rate of 95% or more Li et al. (2020a); Zhong et al. (2020); Li et al. (2021); Wang et al. (2021). Some efforts Zhao et al. (2020); Zhong et al. (2020); Xia et al. (2022a); Zeng et al. (2022) attempt to improve the poisoning efficiency and can be divided into two categories.

**Optimizable Triggers.** How to design an efficient trigger that is easier for DNNs to learn? The existing studies Zhao et al. (2020); Zhong et al. (2020); Zeng et al. (2022) address this issue in terms of the relationship between Universal Adversarial Perturbations (UAPs) Moosavi-Dezfooli et al. (2017) and backdoor triggers, which are highly correlated Pang et al. (2020). On the one hand, a UAP is an inherent flaw in a clean model and can be considered as a natural trigger. On the other hand, when the Trojan injection is complete, the backdoor trigger is actually an attacker-defined UAP for that infected model. Therefore, optimizing a UAP on a pre-trained clean model as the trigger to construct poisoned samples is an effective practice to improve the poisoning efficiency.

**Important Sample Selection.** Xia et al. (2022a) improved the poisoning efficiency by focusing on the selection step. They characterized the learning difficulty of each poisoned sample by recording the number of times it was forgotten during the backdoor injection. In general, poisoned samples with higher forgetting counts are the ones that should be more concerned and contribute more to the backdoor injection. The authors Xia et al. (2022a) confirmed this through data removal and proposed a Filtering-and-Updating Strategy (FUS) to find these high-contribution samples.

### 2.3 THREAT MODEL

Our threat model considers the situation where a user needs to train a DNN model on data scraped from the Internet or provided by a third party. This model is becoming increasingly common as the

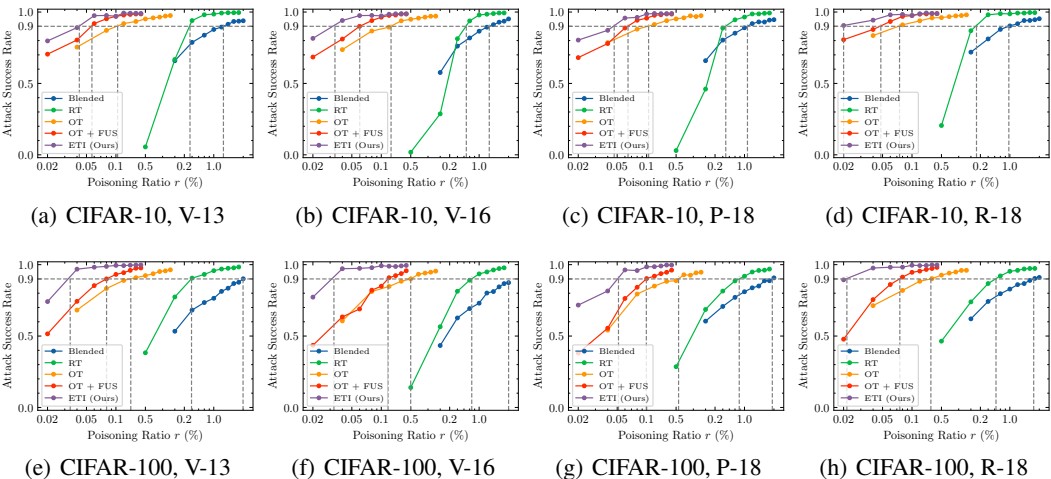

Figure 2: Attack success rates on CIFAR-10 and CIFAR-100, where RT, OT, FUS, and ETI denote Random Trigger, Optimized Trigger, Filtering-and-Updating Strategy, and Efficient Trojan Injection, respectively. Blended Chen et al. (2017) is a common backdoor attack method as a comparison. All curves (except ETI) are averaged over 10 independent runs.

demand for data grows Goldblum et al. (2022). Therefore, we assume that the attacker only has control over the training set: which samples are poisoned and how; he or she has no knowledge of the network architecture and hyperparameters employed by the user.

## 2.4 EXPERIMENTAL SETUP

We try to construct efficient poisoned samples for CIFAR-10 and CIFAR-100 Krizhevsky & Hinton (2009) and specify the attack target $t$ as category 0. Our work is organized into two parts. In the part introducing ETI, to verify its effectiveness, we use VGG-13 (V-13) Simonyan & Zisserman (2014), VGG-16 (V-16) Simonyan & Zisserman (2014), PreActResNet-18 (P-18) He et al. (2016b), and ResNet-18 (R-18) He et al. (2016a) as the DNN architectures and Adam Kingma & Ba (2014) as the optimizer to train the infected models. The total training duration is set to 70, and the batch size is set to 512. The learning rate is initially set to 0.001 and is dropped by 10 after 40 and 60 epochs. It is important to note that since ETI also involves training deep models when generating poisoned samples, here we assume that *the attacker can only use V-13*.

In the second part, we build two backdoored datasets using ETI, namely CIFAR-10-B0-20 and CIFAR-100-B0-30, where only 0.04% (20/50,000) and 0.06% (30/50,000) of the training samples are polluted. To match the threat model, we simulate the user's usage scenario by training the infected models with 10 DNN architectures, 3 optimizers, 4 batch sizes, and 4 initial learning rates. The specific settings can be found in Appendix A. In total, we train 240 models on each dataset to test its attack performance. All experiments are implemented with PyTorch Paszke et al. (2017) and run on an NVIDIA Tesla V100 GPU.

## 3 EFFICIENT TROJAN INJECTION

We now introduce ETI in terms of trigger design, important sample selection, and the exploitation of individual consistency.

### 3.1 OPTIMIZING AN EFFICIENT TRIGGER

A backdoor attack is the process of crafting a loophole for a deep model that causes it to malfunction. The very first step is to create a suitable trigger, either fixed or optimized, which shows great

Table 1: Poisoning ratios $r$ (%) needed to achieve 90% attack success rates on CIFAR-10 using optimized triggers generated with or without image transformations $T(\cdot)$.

|  | V-13 | V-16 | P-18 | R-18 |
|---|---|---|---|---|
| w/ trans. | 0.104 | 0.126 | 0.105 | 0.075 |
| w/o trans. | 0.125 | 0.134 | 0.119 | 0.104 |
| diff. | 0.021 | 0.012 | 0.014 | 0.029 |

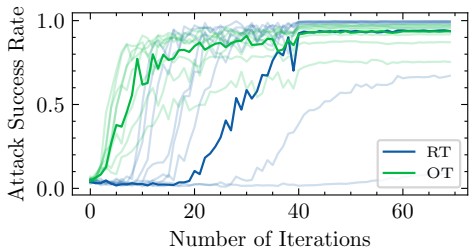

Figure 3: Attack success rate curves using RT and OT on CIFAR-10 and V-13.

importance, especially for poisoning-based backdoor attack methods. But one question arises, *what kind of triggers are the most effective?*

As mentioned in Section 1, in this study, we mainly focus on reducing the number of poisoned samples in the released training set without compromising the effectiveness of the attack. Conceivably, our ultimate goal is to achieve a so-called *zero-shot* backdoor attack, i.e., not poisoning any data during the training phase, but still having a stable and effective trigger. Previous studies Szegedy et al. (2013); Goodfellow et al. (2014); Moosavi-Dezfooli et al. (2017); Hu et al. (2022) have demonstrated that deep models are naturally flawed, so the most intuitive way to implement zero-sample backdoor attacks is to find triggers that can activate these inherent flaws. However, the major problem is that these flaws usually do not qualify as *reliable* model backdoors, manifested in two ways. On the one hand, the best attack success rate that can be achieved falls far short of what we expected. On the other hand, finding an identical defect that can be perfectly applied to all models is difficult. Therefore, it is still very hard to achieve zero-shot backdoor attacks at this moment.

We turn to pursue backdoor attacks with few samples. The discussion above gives a hint: perhaps we can strengthen and consolidate an existing flaw with a small number of poisoned samples to make it eligible as a backdoor, rather than injecting a new one from scratch. Guided by this idea, our attack can be divided into two parts: (1) optimizing a trigger that can activate an inherent flaw in the model, and (2) using this trigger to construct poisoned samples to strengthen the existing flaw. Finding the trigger can be formulated as:

$$\underset{C(t) \leq \epsilon}{\text{minimize}} \sum_{(x,y) \in \mathcal{D}_b} L(f_\theta(T(F(x,t))), y'), \tag{1}$$

where $f_\theta$ denotes a trained benign model and $L(\cdot)$ denotes the loss function. $C(\cdot)$ generally indicates a kind of constraint, while $\epsilon$ defines the upper limit value of $C(t)$. For example, if $C(\cdot)$ represents the area of $t$, the formulation will be reduced to a local pattern-based trigger design. Meanwhile, if $C(\cdot)$ is a norm constraint type, the trigger will cover the whole image, and $\epsilon$ will restrict the pixel changes to ensure the invisibility to some extent. We consider the second case in this study where $C(t) := \|v\|_\infty$, $F(x,t) := x + t$, and $\epsilon = 8/255$.

$T(\cdot)$ is a series of transformations performed on the input image, including random cropping and random flipping. We include the transformations to improve the generalization of the optimized trigger, and the same approach has been shown to be effective in adversarial examples Xie et al. (2019). Table 1 shows the poisoning ratios over 4 different models when achieving 90% attack success rates on CIFAR-10. It can be seen that using triggers generated with transformations requires fewer poisoned samples to reach the same attack strength than using triggers generated without transformations. The ratio is even less than 0.1% on R-18. These results indicate that image transformations help improve the trigger's deformation robustness, thus enhancing its generalization.

Now let us go back to Equation 1. We solve this optimization using the projected gradient descent with the $l_\infty$-norm constraint Madry et al. (2017), which updates the trigger $t$ along the direction of the gradient sign for multiple iterations. The detailed algorithm is given in Appendix B.

To verify the effectiveness of the triggers devised from the above technique, we implant backdoors on 4 DNN models with different poisoning ratios on CIFAR-10 and CIFAR-100, and the attack success rates are shown in Figure 2. As a comparison, we also test the attack performance when

using Randomly generated perturbations as Triggers (RT) under the same constraint, i.e., $l_\infty$-norm and $\epsilon = 8/255$. It can be seen that the poisoning ratio required for the Optimized Trigger (OT) is much lower than that of RT for obtaining the same attack strength. Concretely, when the attack success rate reaches 90%, we need to poison 0.103% and 0.178% using OT on CIFAR-10 and CIFAR-100, respectively, whereas 0.603% and 0.761% are required regarding RT.

In addition, we plot the learning curves of injection on CIFAR-10 using RT and OT at different poisoning ratios, as shown in Figure 3. Note that the number of poisoned samples ranges from 20 to 180 for OT and from 100 to 900 for RT, respectively. The green and blue lines highlighted are the results of 80 and 300. As can be seen, in spite of the similar final values after convergence, the learning processes are completely distinct. Specifically, neural networks manage to learn the features of OT at the beginning of the training, but gradually gain information of RT only after several epochs. This observation explains to some extent why OT is more efficient: strengthening the inherent flaw and learning the decision of the original task overlap considerably in direction.

## 3.2 SELECTING IMPORTANT SAMPLES

After optimizing an efficient trigger, picking which benign samples to poison is also an essential step. In almost all previous work, the samples to be poisoned are chosen randomly, based on the assumption that each adversary contributes equally to the backdoor injection. But that is not how it works. In regular classification tasks, several studies Katharopoulos & Fleuret (2018); Toneva et al. (2018) have shown that some hard or forgettable samples are more important for forming the decisions of DNNs. Recently, Xia et al. (2022a) suggested that forgettable poisoned samples – whose predictions are prone to change during the training cycle – are more significant than unforgettable ones with regard to the poisoning efficiency. We agree with their conclusion and apply the algorithm named FUS proposed in that paper to further satisfy our need for a smaller amount of poisoned data. The detailed algorithm of FUS is given in Appendix C.

The main idea of FUS is to find poisoned samples with large forgetting events by filtering and updating the sample pool. This process is usually iterated 10 to 15 times, and the last index of selected samples is saved. The specific algorithm can be found in Xia et al. (2022a). Through simple experiments, we find that this algorithm is indeed effective in promoting sample efficiency. However, randomness becomes influential on the FUS outcome when the poisoning ratio plummets to a fairly small amount, such as only a few tens of poisoned samples (less than 0.1%). We conduct a validation experiment to describe this impact, and the results are shown in Figure 4. The blue dots represent the average number of iterations when the best backdoor attack accuracy occurs, and the green dots provide the average difference between the best and last attack success rates. We can see that if the poisoning ratio is set to 0.02% (merely 10 samples), the best result is obtained

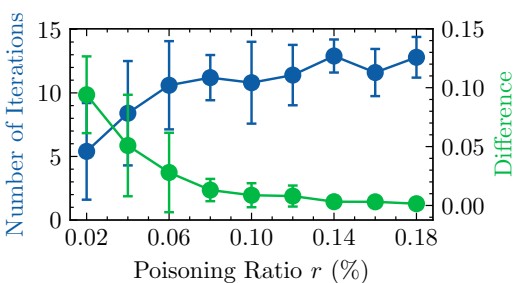

Figure 4: Experimental results of FUS on CIFAR-10 at very small poisoning ratios. We set the total number of iterations of the algorithm to 15. The blue line is the number of iterations to achieve the best attack, and the green line is the difference between the best attack success rate and the last attack success rate achieved.

at around the 5th iteration, and the difference is very large. But as the ratio increases, the difference gradually decreases to 0, which means that the last result is almost equal to the best result.

Based on the above observation, we make a simple improvement to the original FUS algorithm for small size of poisoned samples, that is, to save the best instead of the last sample index result. Next, we combine FUS with Optimized Trigger (OT) generated in Section 3.1 to obtain the corresponding curves in Figure 2. It is apparent that the poisoning ratios required to reach 90% attack success rates decrease further, with the exact numbers dropping to 0.058% and 0.093% on CIFAR-10 and CIFAR-100, respectively.

Table 2: Poisoning ratios $r$ (%) needed to achieve 90% attack success rates for the poisoned sample sets generated from 10 independent runs using OT + FUS

|       | 0     | 1     | 2     | 3     | 4     | 5     | 6     | 7     | 8     | 9     |
|-------|-------|-------|-------|-------|-------|-------|-------|-------|-------|-------|
| V-13  | 0.069 | 0.058 | 0.053 | 0.042 | 0.054 | 0.069 | 0.067 | 0.047 | 0.053 | 0.058 |
| V-16  | 0.074 | 0.075 | 0.056 | 0.034 | 0.047 | 0.068 | 0.072 | 0.041 | 0.068 | 0.068 |
| P-18  | 0.070 | 0.082 | 0.057 | 0.047 | 0.070 | 0.073 | 0.069 | 0.047 | 0.065 | 0.056 |
| R-18  | 0.058 | 0.059 | 0.050 | 0.020 | 0.039 | 0.045 | 0.053 | 0.039 | 0.032 | 0.054 |
| Mean  | 0.068 | 0.069 | 0.054 | **0.036** | 0.053 | 0.064 | 0.065 | 0.044 | 0.055 | 0.059 |

### 3.3 UTILIZING THE INDIVIDUAL CONSISTENCY

In the first two parts, we design optimized triggers that exploit the natural flaws of deep models, as well as select tens of samples that contribute more to the backdoor injection process. Since these techniques involve randomness, e.g., the initial sample pool in FUS is randomly sampled, the results above are the average of 10 independent runs. Statistical analysis is beneficial when comparing different approaches; however, when practically injecting a Trojan, we usually need to focus on a specific individual. Our concern in this section is to analyze the differences between *individuals*.

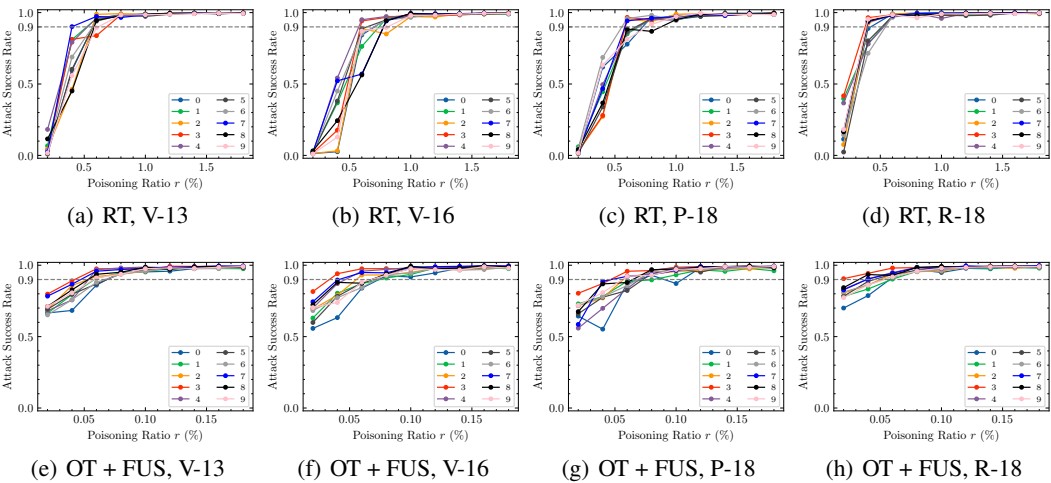

(a) RT, V-13      (b) RT, V-16      (c) RT, P-18      (d) RT, R-18

(e) OT + FUS, V-13      (f) OT + FUS, V-16      (g) OT + FUS, P-18      (h) OT + FUS, R-18

Figure 5: Attack success rates for the poisoned sample sets generated from 10 independent runs using RT and OT + FUS on CIFAR-10.

Some experiments are conducted to observe the performance of individuals. First, we define the poisoned sample set $\mathcal{D}_p$ created by OT and FUS as an individual. Next, we run these two techniques independently 10 times to obtain $\mathcal{D}_p^0, \mathcal{D}_p^1, \cdots, \mathcal{D}_p^9$. Last, we use these 10 individuals for poisoning and train 4 models to evaluate their attack performance. The corresponding results on CIFAR-10 are shown in Figure 5 and Table 2. As a comparison, we do the same experiments with RT and RSS. The results on CIFAR-100 are similar and can be seen in Appendix D.

From the above figures, the existence of individual differences can be demonstrated. We can observe that with the same number of poisoned data, the attack success rates are reported differently for different runs, even by a factor of several if the poisoning ratio is minor. Despite this common feature, we identify a unique characteristic only for OT, namely that the performance of these individuals exhibits a great consistency across models. For example, in the OT results in Figure 5, the red line (run 3) is always on the top, while the blue line (run 0) performs the worst all the time. Other lines, such as green and pink, basically show moderate performance. However, we can not find such correlations from the RT results.

Besides, to quantify this individual consistency, we perform Pearson correlation analysis of the 4 models on both RT and OT + FUS, and the p-value matrices and correlation coefficient matrices are shown in Figure 6. For OT + FUS, almost all p-values are below 0.05, and the correlation coefficients are above 0.7, revealing that there are significant and positive correlations between the models. In contrast, for RT, the p-values become fairly large, and the coefficients decline to about 0.15 or even turn negative.

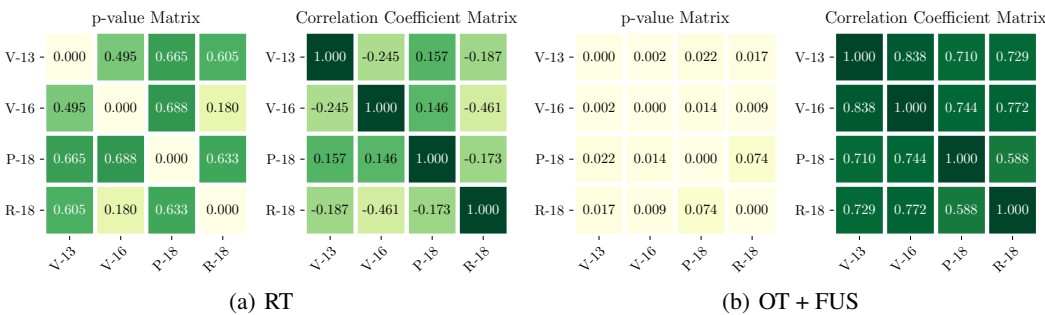

Figure 6: Pearson correlation analysis on both RT and OT + FUS on CIFAR-10.

Overall, these results collectively provide an important insight to the individual consistency of OT + FUS, i.e., if an individual performs well on one model, it is more likely to achieve high accuracy on others. As shown in the corresponding lines (ETI) in Figure 2, we use this characteristic to select an individual with the best performance and further reduce the poisoning ratios to 0.036% and 0.035% on CIFAR-10 and CIFAR-100, respectively.

## 4 CIFAR-10-B0-20 AND CIFAR-100-B0-30

In this section, we use ETI to build two datasets named CIFAR-10-B0-20 and CIFAR-100-B0-30, corresponding to the backdoored versions of CIFAR-10 and CIFAR-100, respectively. In the names, "B" represents "Backdoor", "0" represents that the attack target $y'$ is set to category 0, and "20" or "30" represent the number of poisoned samples. The poisoned images in CIFAR-10-B0-20 are shown in Figure 7. As we can see, these images maintain a large visual similarity to the original ones. In the same way, we list the poisoning images in CIFAR-100-B0-30, see Appendix D.

We test the attack performance of CIFAR-10-B0-20 on 240 models with different network structures and training hyperparameters, recorded in Table 3. The average attack success rate that can be achieved using 20 poisoned samples on CIFAR-10 is 92.1%. 78 out of 240 models are greater than 95%, and 198 out of 240 models are greater than 90%. However, the poisoning is not always successful either. For example, 10 out of 240 models are less than 80%, accounting for approximately 4.2%. We even got a 25.8% attack success rate in row 19, column 1, where ResNet-50, SGD optimizer, initial learning rate 0.03 and batch size 512 are used. Similarly, as to CIFAR-100-B0-30, we achieve an average backdoor accuracy of 90.4%. 201 out of 240 models are greater than 90%, yet 27 out of 240 are less than 80%. The detailed results are shown in Appendix E.

We also constructed CIFAR-10-B0-30 and CIFAR-100-B0-40, with 10 more poisoned samples, and the results can be seen in Appendix F. In these cases, we achieve an average attack success rate of 95.2% and 95.1%, respectively, where only 4 and 13 out of 240 models are less than 80%.

Taken together, these results above indicate that it is practical to achieve a high success rate by contaminating only a few tens of samples out of 50,000 clean data, without access to the structure and hyperparameters of the model used by the user.

## 5 CONCLUSION AND FUTURE WORK

Our study illustrates that there is still great potential for data-efficient backdoor attacks. We achieve over 90% attack success rates on CIFAR-10 and CIFAR-100 with just 0.04% and 0.06% poisoned

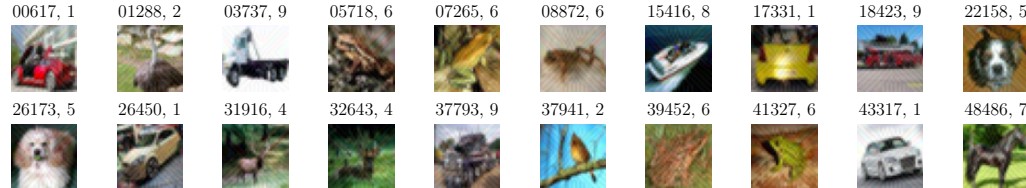

Figure 7: Poisoned samples in CIFAR-10-B0-20. The two numbers above each image represent its sequential position in the training set and its original label, respectively.

Table 3: Attack success rates of models trained on CIFAR-10-B0-20. The horizontal numbers represent the numbering of different DNN architectures and the vertical numbers represent the numbering of different training hyperparameters. See Appendix A for the specific meaning of each number.

|    | 0 | 1 | 2 | 3 | 4 | 5 | 6 | 7 | 8 | 9 |
|----|-------|-------|-------|-------|-------|-------|-------|-------|-------|-------|
| 0  | 0.959 | 0.972 | 0.965 | 0.982 | 0.981 | 0.951 | 0.937 | 0.908 | 0.974 | 0.934 |
| 1  | 0.971 | 0.973 | 0.961 | 0.984 | 0.974 | 0.973 | 0.951 | 0.936 | 0.965 | 0.931 |
| 2  | 0.952 | 0.907 | 0.924 | 0.886 | 0.939 | 0.877 | 0.949 | 0.950 | 0.950 | 0.922 |
| 3  | 0.958 | 0.772 | 0.900 | 0.927 | 0.885 | 0.898 | 0.928 | 0.890 | 0.953 | 0.875 |
| 4  | 0.938 | 0.948 | 0.945 | 0.960 | 0.965 | 0.979 | 0.928 | 0.928 | 0.956 | 0.938 |
| 5  | 0.968 | 0.931 | 0.888 | 0.955 | 0.969 | 0.968 | 0.944 | 0.905 | 0.949 | 0.928 |
| 6  | 0.944 | 0.949 | 0.930 | 0.969 | 0.966 | 0.969 | 0.954 | 0.918 | 0.950 | 0.941 |
| 7  | 0.971 | 0.968 | 0.959 | 0.976 | 0.964 | 0.964 | 0.945 | 0.902 | 0.966 | 0.935 |
| 8  | 0.960 | 0.932 | 0.906 | 0.955 | 0.969 | 0.952 | 0.959 | 0.929 | 0.953 | 0.911 |
| 9  | 0.952 | 0.912 | 0.884 | 0.914 | 0.939 | 0.923 | 0.917 | 0.937 | 0.946 | 0.954 |
| 10 | 0.930 | 0.948 | 0.886 | 0.940 | 0.972 | 0.968 | 0.903 | 0.908 | 0.941 | 0.856 |
| 11 | 0.926 | 0.944 | 0.796 | 0.939 | 0.942 | 0.948 | 0.913 | 0.928 | 0.961 | 0.897 |
| 12 | 0.918 | 0.949 | 0.800 | 0.961 | 0.937 | 0.962 | 0.898 | 0.924 | 0.956 | 0.949 |
| 13 | 0.949 | 0.967 | 0.611 | 0.968 | 0.960 | 0.969 | 0.929 | 0.926 | 0.956 | 0.934 |
| 14 | 0.952 | 0.951 | 0.926 | 0.945 | 0.977 | 0.964 | 0.952 | 0.934 | 0.940 | 0.910 |
| 15 | 0.935 | 0.959 | 0.852 | 0.946 | 0.931 | 0.946 | 0.946 | 0.944 | 0.947 | 0.922 |
| 16 | 0.943 | 0.966 | 0.843 | 0.963 | 0.934 | 0.947 | 0.817 | 0.863 | 0.930 | 0.865 |
| 17 | 0.938 | 0.953 | 0.754 | 0.949 | 0.916 | 0.947 | 0.908 | 0.937 | 0.929 | 0.922 |
| 18 | 0.818 | 0.827 | 0.569 | 0.895 | 0.926 | 0.926 | 0.918 | 0.870 | 0.910 | 0.840 |
| 19 | 0.930 | 0.258 | 0.578 | 0.876 | 0.948 | 0.953 | 0.856 | 0.899 | 0.944 | 0.924 |
| 20 | 0.960 | 0.934 | 0.904 | 0.945 | 0.962 | 0.973 | 0.925 | 0.919 | 0.938 | 0.860 |
| 21 | 0.936 | 0.956 | 0.904 | 0.964 | 0.965 | 0.920 | 0.938 | 0.928 | 0.955 | 0.923 |
| 22 | 0.934 | 0.938 | 0.884 | 0.932 | 0.921 | 0.958 | 0.792 | 0.875 | 0.916 | 0.640 |
| 23 | 0.934 | 0.954 | 0.588 | 0.946 | 0.876 | 0.972 | 0.925 | 0.915 | 0.937 | 0.893 |

Mean: 0.921, STD: 0.074

samples through efficient trigger design, important sample selection, and the exploitation of individual consistency. It is a considerable improvement compared to previous work.

Much work remains to be explored in the future. The first is the pursuit of more extreme sample volumes. Is it possible to complete a backdoor attack with a few samples, or even one sample? Second, our experimental results show lower attack success rates under some training conditions. Why do these situations occur? How to enhance the generalization of the poisoned samples? Third, We focus on the most basic image classification tasks, what form should efficient poisoned samples take for other tasks? And so on.

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

# A DNN ARCHITECTURES AND TRAINING HYPERPARAMETERS

10 DNN architectures and 24 training hyperparameters that we use in testing the poisoning performance of CIFAR-10-B0-20 and CIFAR-100-B0-30 are shown in Table 4 and Table 5. It should be noted that the attacker builds these datasets using only VGG-13.

Table 4: 10 DNN Architectures.

| No. | Model Name | No. | Model Name |
|---|---|---|---|
| 0 | ResNet-34 He et al. (2016a) | 1 | ResNet-50 He et al. (2016a) |
| 2 | PreActResNet-34He et al. (2016b) | 3 | PreActResNet-50 He et al. (2016b) |
| 4 | SENet-18 Hu et al. (2018) | 5 | ResNeXt-29 (2x64d) Xie et al. (2017) |
| 6 | RegNetX-200MF Radosavovic et al. (2020) | 7 | MobileNetV2 Sandler et al. (2018) |
| 8 | DenseNet-121 Huang et al. (2017) | 9 | EfficientNet-B0 Tan & Le (2019) |

Table 5: 24 training hyperparameters. OPT: Optimizer. ILR: Initial Learning Rate. BS: Batch Size.

| No. | OPT | ILR | BS | No. | OPT | ILR | BS |
|---|---|---|---|---|---|---|---|
| 0 | SGD | 0.02 | 64 | 12 | SGD | 0.02 | 256 |
| 1 | SGD | 0.03 | 64 | 13 | SGD | 0.03 | 256 |
| 2 | Adam | 0.001 | 64 | 14 | Adam | 0.001 | 256 |
| 3 | Adam | 0.002 | 64 | 15 | Adam | 0.002 | 256 |
| 4 | AdamW | 0.001 | 64 | 16 | AdamW | 0.001 | 256 |
| 5 | AdamW | 0.002 | 64 | 17 | AdamW | 0.002 | 256 |
| 6 | SGD | 0.02 | 128 | 18 | SGD | 0.02 | 512 |
| 7 | SGD | 0.03 | 128 | 19 | SGD | 0.03 | 512 |
| 8 | Adam | 0.001 | 128 | 20 | Adam | 0.001 | 512 |
| 9 | Adam | 0.002 | 128 | 21 | Adam | 0.002 | 512 |
| 10 | AdamW | 0.001 | 128 | 22 | AdamW | 0.001 | 512 |
| 11 | AdamW | 0.002 | 128 | 23 | AdamW | 0.002 | 512 |

# B  OPTIMIZED TRIGGER GENERATION ALGORITHM

The optimized trigger generation algorithm is shown in Algorithm 1. In this study, we set $N_{ot} = 300$.

---

**Algorithm 1:** Optimized Trigger Generation Algorithm

---

**Input:** Number of iterations $N_{ot}$; Benign training dataset $\mathcal{D}_b$; Attack target $y'$; Fusion fuction
$F$; Input transformation $T$; Clean pretrained model $f_\theta$; Trigger constraint $C$; Bound
value $\epsilon$; Step size $\alpha$

**Output:** Optimized trigger $t$

Sample random initial value $t \sim U(-1, 1)$, with $C(t) \leq \epsilon$;

**for** $n \leftarrow 1$ **to** $N_{ot}$ **do**
    **for** $X \in \mathcal{D}_b$ **do**
        $\eta = \text{sign}(\nabla L(f_\theta(T(F(X, t))), y'))$;
        $t = t - \alpha \cdot \eta$, with $C(t) \leq \epsilon$;
    **end**
**end**

---

## C  FILTERING-AND-UPDATING STRATEGY

The procedure of FUS is shown in Algorithm 2. In this study, we set $N_{fus} = 15$ and $\alpha = 0.2$.

---

**Algorithm 2:** Filtering-and-Updating Strategy

---

**Input:** Number of iterations $N_{fus}$; Benign training dataset $\mathcal{D}_b$; Attack target $y'$; Fusion fuction $F$; Backdoor trigger $t$; Poisoning ratio $r$; Filtration ratio $\alpha$

**Output:** Constructed poisoned training set $\mathcal{D}_p$

Initialize the sample pool $\mathcal{D}'_s$ by randomly sampling $r \cdot |\mathcal{D}|$ samples from $\mathcal{D}_b$;

**for** $n \leftarrow 1$ **to** $N_{fus}$ **do**

    **Filtering step:**

        Build the corresponding poisoned set $\mathcal{D}'_p = \{(F(x,t), y') | (x,y) \in \mathcal{D}'_s\}$;

        Train an infected model $f_\theta$ from scratch on $\mathcal{D}_m = (\mathcal{D}_b \setminus \mathcal{D}'_s) \cup \mathcal{D}'_p$, and record the forgetting events for each sample in $\mathcal{D}'_p$;

        Filter out $\alpha \cdot r \cdot |\mathcal{D}|$ samples in $\mathcal{D}'_s$ according to the order of corresponding forgetting events from small to large in $\mathcal{D}'_p$;

    **Updating step:**

        Update $\mathcal{D}'_s$ by randomly sampling $\alpha \cdot r \cdot |\mathcal{D}|$ samples from $\mathcal{D}_b$ and adding to the sample pool;

**end**

Return the constructed poisoned sample set $\mathcal{D}_p = \{(F(x,t), y') | (x,y) \in \mathcal{D}'_s\}$

---

# D INDIVIDUAL DIFFERENCES AND CONSISTENCY ANALYSIS ON CIFAR-100

We similarly perform the individual differences and consistency analysis on CIFAR-100. The results of 10 runs on CIFAR-100 are shown in Figure 8 and Table 6. We perform Pearson correlation analysis of the 4 models on both RT and OT + FUS, and the p-value matrices and correlation coefficient matrices are shown in Figure 9. For OT + FUS, all p-values are below 0.05, and the correlation coefficients are above 0.88, revealing that there are significant and positive correlations between the models. In contrast, for RT, the p-values become fairly large, and the coefficients decline to about 0 or even turn negative.

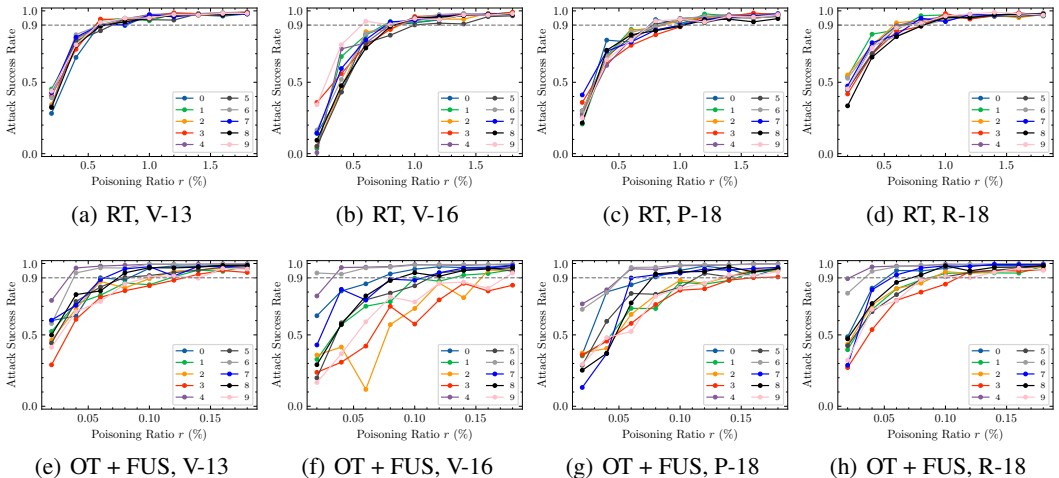

Figure 8: Attack success rates for the poisoned sample sets generated from 10 independent runs using RT and OT + FUS on CIFAR-100.

Table 6: Poisoning ratios $r$ (%) needed to achieve 90% attack success rates for the poisoned sample sets generated from 10 independent runs using OT + FUS on CIFAR-100.

|        | 0     | 1     | 2     | 3     | 4         | 5     | 6     | 7     | 8     | 9     |
|--------|-------|-------|-------|-------|-----------|-------|-------|-------|-------|-------|
| V-13   | 0.084 | 0.118 | 0.102 | 0.128 | 0.034     | 0.079 | 0.038 | 0.063 | 0.075 | 0.097 |
| V-16   | 0.072 | 0.131 | 0.155 | 0.180 | 0.033     | 0.113 | 0.020 | 0.107 | 0.087 | 0.174 |
| P-18   | 0.078 | 0.146 | 0.137 | 0.159 | 0.052     | 0.114 | 0.052 | 0.061 | 0.077 | 0.133 |
| R-18   | 0.052 | 0.087 | 0.090 | 0.110 | 0.022     | 0.093 | 0.034 | 0.056 | 0.072 | 0.092 |
| Mean   | 0.072 | 0.121 | 0.121 | 0.144 | **0.035** | 0.100 | 0.036 | 0.072 | 0.078 | 0.124 |

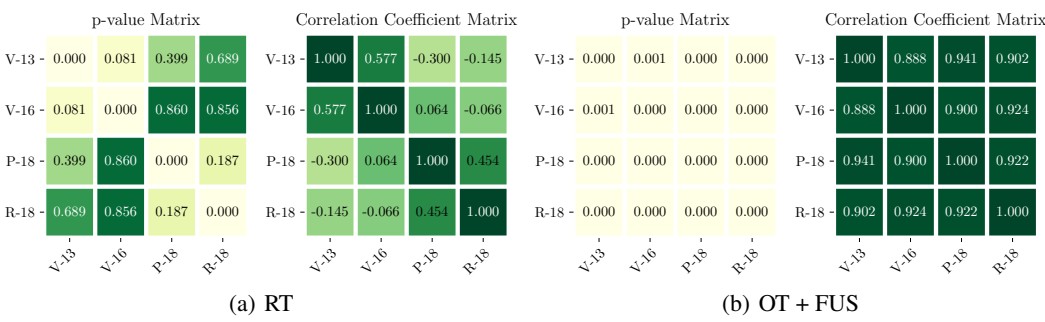

Figure 9: Pearson correlation analysis on both RT and OT + FUS on CIFAR-100.

# E CIFAR-100-30-B0

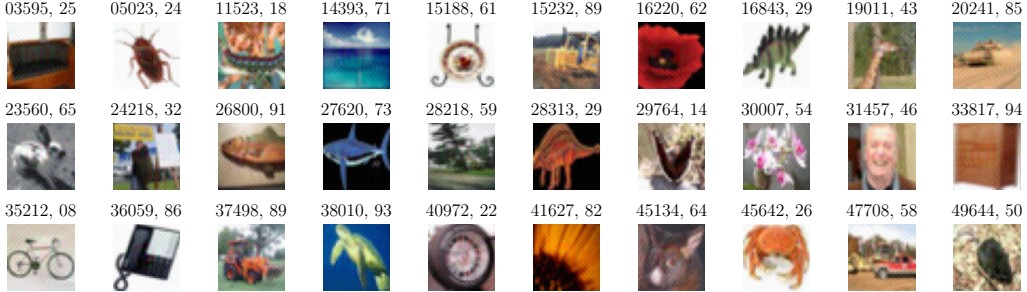

Figure 10: Poisoned samples in CIFAR-100-B0-30. The two numbers above each image represent its sequential position in the training set and its original label, respectively.

Table 7: Attack success rates of models trained on CIFAR-100-B0-30. The horizontal numbers represent the numbering of different network structures and the vertical numbers represent the numbering of different training hyperparameters.

|    | 0     | 1     | 2     | 3     | 4     | 5     | 6     | 7     | 8     | 9     |
|----|-------|-------|-------|-------|-------|-------|-------|-------|-------|-------|
| 0  | 0.996 | 0.996 | 0.997 | 0.997 | 0.995 | 0.994 | 0.994 | 0.997 | 0.996 | 0.859 |
| 1  | 0.997 | 0.995 | 0.996 | 0.996 | 0.996 | 0.997 | 0.998 | 0.990 | 0.997 | 0.984 |
| 2  | 0.994 | 0.986 | 0.993 | 0.978 | 0.997 | 0.988 | 0.993 | 0.985 | 0.994 | 0.572 |
| 3  | 0.987 | 0.979 | 0.979 | 0.981 | 0.990 | 0.987 | 0.995 | 0.984 | 0.983 | 0.933 |
| 4  | 0.944 | 0.992 | 0.004 | 0.981 | 0.615 | 0.998 | 0.978 | 0.958 | 0.973 | 0.956 |
| 5  | 0.901 | 0.966 | 0.186 | 0.954 | 0.075 | 0.994 | 0.986 | 0.990 | 0.940 | 0.982 |
| 6  | 0.986 | 0.993 | 0.989 | 0.993 | 0.985 | 0.993 | 0.985 | 0.992 | 0.992 | 0.976 |
| 7  | 0.991 | 0.994 | 0.991 | 0.998 | 0.991 | 0.993 | 0.995 | 0.987 | 0.997 | 0.958 |
| 8  | 0.988 | 0.979 | 0.991 | 0.987 | 0.985 | 0.985 | 0.983 | 0.977 | 0.992 | 0.811 |
| 9  | 0.995 | 0.987 | 0.989 | 0.962 | 0.962 | 0.995 | 0.989 | 0.973 | 0.984 | 0.822 |
| 10 | 0.735 | 0.983 | 0.014 | 0.961 | 0.635 | 0.992 | 0.958 | 0.963 | 0.975 | 0.908 |
| 11 | 0.300 | 0.972 | 0.162 | 0.955 | 0.018 | 0.977 | 0.967 | 0.949 | 0.988 | 0.897 |
| 12 | 0.491 | 0.987 | 0.961 | 0.967 | 0.941 | 0.984 | 0.960 | 0.972 | 0.976 | 0.894 |
| 13 | 0.982 | 0.983 | 0.988 | 0.990 | 0.983 | 0.988 | 0.971 | 0.988 | 0.982 | 0.952 |
| 14 | 0.978 | 0.992 | 0.964 | 0.976 | 0.988 | 0.994 | 0.974 | 0.969 | 0.986 | 0.978 |
| 15 | 0.989 | 0.986 | 0.987 | 0.977 | 0.983 | 0.977 | 0.992 | 0.985 | 0.982 | 0.936 |
| 16 | 0.915 | 0.988 | 0.666 | 0.964 | 0.956 | 0.988 | 0.886 | 0.956 | 0.964 | 0.339 |
| 17 | 0.682 | 0.974 | 0.013 | 0.972 | 0.539 | 0.979 | 0.969 | 0.964 | 0.958 | 0.956 |
| 18 | 0.082 | 0.325 | 0.684 | 0.866 | 0.838 | 0.976 | 0.766 | 0.960 | 0.963 | 0.956 |
| 19 | 0.971 | 0.939 | 0.939 | 0.958 | 0.929 | 0.974 | 0.931 | 0.954 | 0.960 | 0.970 |
| 20 | 0.986 | 0.984 | 0.945 | 0.982 | 0.988 | 0.990 | 0.943 | 0.971 | 0.943 | 0.890 |
| 21 | 0.979 | 0.979 | 0.838 | 0.960 | 0.982 | 0.984 | 0.980 | 0.971 | 0.981 | 0.951 |
| 22 | 0.927 | 0.977 | 0.083 | 0.905 | 0.918 | 0.983 | 0.857 | 0.944 | 0.832 | 0.686 |
| 23 | 0.303 | 0.969 | 0.349 | 0.698 | 0.576 | 0.975 | 0.915 | 0.949 | 0.971 | 0.929 |

Mean: 0.904, STD: 0.205

# F   CIFAR-10-B0-30 AND CIFAR-100-B0-40

Table 8: Attack success rates of models trained on CIFAR-10-B0-30. The horizontal numbers represent the numbering of different network structures and the vertical numbers represent the numbering of different training hyperparameters.

|    | 0 | 1 | 2 | 3 | 4 | 5 | 6 | 7 | 8 | 9 |
|----|-------|-------|-------|-------|-------|-------|-------|-------|-------|-------|
| 0  | 0.981 | 0.983 | 0.970 | 0.984 | 0.986 | 0.983 | 0.984 | 0.961 | 0.978 | 0.978 |
| 1  | 0.987 | 0.987 | 0.978 | 0.983 | 0.988 | 0.986 | 0.976 | 0.967 | 0.980 | 0.969 |
| 2  | 0.977 | 0.955 | 0.955 | 0.972 | 0.980 | 0.938 | 0.958 | 0.967 | 0.974 | 0.937 |
| 3  | 0.964 | 0.918 | 0.932 | 0.861 | 0.944 | 0.874 | 0.951 | 0.947 | 0.978 | 0.861 |
| 4  | 0.970 | 0.980 | 0.959 | 0.985 | 0.985 | 0.979 | 0.970 | 0.958 | 0.966 | 0.928 |
| 5  | 0.983 | 0.968 | 0.978 | 0.974 | 0.971 | 0.981 | 0.946 | 0.967 | 0.972 | 0.959 |
| 6  | 0.980 | 0.978 | 0.894 | 0.979 | 0.984 | 0.979 | 0.970 | 0.964 | 0.986 | 0.962 |
| 7  | 0.976 | 0.972 | 0.963 | 0.981 | 0.989 | 0.977 | 0.972 | 0.953 | 0.982 | 0.971 |
| 8  | 0.980 | 0.970 | 0.976 | 0.951 | 0.967 | 0.978 | 0.964 | 0.965 | 0.964 | 0.958 |
| 9  | 0.942 | 0.940 | 0.952 | 0.907 | 0.951 | 0.940 | 0.943 | 0.932 | 0.955 | 0.964 |
| 10 | 0.971 | 0.971 | 0.929 | 0.977 | 0.968 | 0.971 | 0.939 | 0.950 | 0.979 | 0.952 |
| 11 | 0.972 | 0.979 | 0.920 | 0.981 | 0.974 | 0.969 | 0.946 | 0.965 | 0.968 | 0.958 |
| 12 | 0.965 | 0.919 | 0.930 | 0.970 | 0.987 | 0.978 | 0.967 | 0.960 | 0.975 | 0.960 |
| 13 | 0.966 | 0.950 | 0.728 | 0.973 | 0.981 | 0.977 | 0.953 | 0.953 | 0.968 | 0.951 |
| 14 | 0.976 | 0.979 | 0.971 | 0.958 | 0.972 | 0.983 | 0.955 | 0.967 | 0.977 | 0.927 |
| 15 | 0.950 | 0.970 | 0.963 | 0.952 | 0.983 | 0.974 | 0.956 | 0.958 | 0.962 | 0.955 |
| 16 | 0.975 | 0.967 | 0.947 | 0.970 | 0.968 | 0.978 | 0.903 | 0.922 | 0.966 | 0.893 |
| 17 | 0.932 | 0.977 | 0.873 | 0.974 | 0.980 | 0.957 | 0.956 | 0.954 | 0.970 | 0.968 |
| 18 | 0.942 | 0.893 | 0.799 | 0.960 | 0.959 | 0.968 | 0.926 | 0.912 | 0.955 | 0.855 |
| 19 | 0.964 | 0.955 | 0.064 | 0.972 | 0.975 | 0.973 | 0.949 | 0.944 | 0.956 | 0.913 |
| 20 | 0.977 | 0.966 | 0.954 | 0.954 | 0.980 | 0.972 | 0.943 | 0.950 | 0.959 | 0.957 |
| 21 | 0.970 | 0.947 | 0.953 | 0.968 | 0.983 | 0.957 | 0.965 | 0.972 | 0.976 | 0.962 |
| 22 | 0.967 | 0.968 | 0.881 | 0.980 | 0.972 | 0.983 | 0.871 | 0.927 | 0.925 | 0.785 |
| 23 | 0.941 | 0.952 | 0.832 | 0.966 | 0.962 | 0.981 | 0.920 | 0.954 | 0.936 | 0.907 |

Mean: 0.952, STD: 0.067

Table 9: Attack success rates of models trained on CIFAR-100-B0-40. The horizontal numbers represent the numbering of different network structures and the vertical numbers represent the numbering of different training hyperparameters.

|    | 0 | 1 | 2 | 3 | 4 | 5 | 6 | 7 | 8 | 9 |
|----|-------|-------|-------|-------|-------|-------|-------|-------|-------|-------|
| 0  | 0.998 | 0.998 | 0.998 | 0.998 | 0.997 | 0.997 | 0.994 | 0.996 | 0.995 | 0.816 |
| 1  | 0.999 | 0.998 | 0.999 | 0.996 | 0.984 | 0.994 | 0.998 | 0.996 | 0.999 | 0.954 |
| 2  | 0.998 | 0.993 | 0.996 | 0.990 | 0.997 | 0.997 | 0.992 | 0.997 | 0.996 | 0.902 |
| 3  | 0.983 | 0.974 | 0.995 | 0.996 | 0.998 | 0.982 | 0.997 | 0.992 | 0.993 | 0.686 |
| 4  | 0.939 | 0.997 | 0.003 | 0.983 | 0.948 | 0.992 | 0.990 | 0.986 | 0.987 | 0.977 |
| 5  | 0.679 | 0.981 | 0.975 | 0.982 | 0.034 | 0.990 | 0.993 | 0.989 | 0.989 | 0.987 |
| 6  | 0.995 | 0.996 | 0.997 | 0.997 | 0.996 | 0.994 | 0.993 | 0.993 | 0.992 | 0.928 |
| 7  | 0.997 | 0.996 | 0.999 | 0.998 | 0.991 | 0.997 | 0.995 | 0.992 | 0.997 | 0.993 |
| 8  | 0.998 | 0.998 | 0.995 | 0.991 | 0.998 | 0.992 | 0.992 | 0.995 | 0.994 | 0.937 |
| 9  | 0.993 | 0.993 | 0.995 | 0.992 | 0.994 | 0.996 | 0.984 | 0.974 | 0.993 | 0.949 |
| 10 | 0.977 | 0.994 | 0.941 | 0.984 | 0.974 | 0.992 | 0.974 | 0.988 | 0.991 | 0.952 |
| 11 | 0.704 | 0.989 | 0.928 | 0.986 | 0.980 | 0.992 | 0.983 | 0.971 | 0.991 | 0.983 |
| 12 | 0.983 | 0.985 | 0.982 | 0.994 | 0.970 | 0.993 | 0.986 | 0.993 | 0.981 | 0.971 |
| 13 | 0.985 | 0.992 | 0.990 | 0.997 | 0.995 | 0.995 | 0.989 | 0.990 | 0.992 | 0.992 |
| 14 | 0.994 | 0.995 | 0.993 | 0.991 | 0.992 | 0.991 | 0.989 | 0.997 | 0.992 | 0.978 |
| 15 | 0.990 | 0.996 | 0.984 | 0.993 | 0.997 | 0.987 | 0.992 | 0.997 | 0.990 | 0.975 |
| 16 | 0.627 | 0.992 | 0.946 | 0.980 | 0.974 | 0.988 | 0.950 | 0.974 | 0.982 | 0.969 |
| 17 | 0.947 | 0.986 | 0.027 | 0.979 | 0.800 | 0.989 | 0.986 | 0.981 | 0.977 | 0.959 |
| 18 | 0.950 | 0.217 | 0.957 | 0.918 | 0.967 | 0.986 | 0.950 | 0.976 | 0.983 | 0.364 |
| 19 | 0.971 | 0.722 | 0.945 | 0.988 | 0.977 | 0.993 | 0.954 | 0.982 | 0.984 | 0.960 |
| 20 | 0.993 | 0.993 | 0.988 | 0.989 | 0.979 | 0.998 | 0.977 | 0.979 | 0.977 | 0.974 |
| 21 | 0.978 | 0.981 | 0.963 | 0.986 | 0.990 | 0.993 | 0.986 | 0.990 | 0.991 | 0.972 |
| 22 | 0.951 | 0.985 | 0.599 | 0.975 | 0.989 | 0.993 | 0.950 | 0.946 | 0.912 | 0.935 |
| 23 | 0.659 | 0.980 | 0.913 | 0.980 | 0.296 | 0.995 | 0.947 | 0.977 | 0.967 | 0.922 |

Mean: 0.951, STD: 0.142

# G CLEAN ACCURACY ON CIFAR-10 AND CIFAR-10-B0-20

Clean accuracy of 240 models trained on CIFAR-10 and CIFAR-10-B0-20 are shown in Table 10 and Table 11, respectively. It can be seen that the ETI-generated poisoned samples have almost no effect on the clean accuracy.

Table 10: Clean accuracy of models trained on CIFAR-10. The horizontal numbers represent the numbering of different network structures and the vertical numbers represent the numbering of different training hyperparameters.

|    | 0 | 1 | 2 | 3 | 4 | 5 | 6 | 7 | 8 | 9 |
|----|-------|-------|-------|-------|-------|-------|-------|-------|-------|-------|
| 0  | 0.951 | 0.948 | 0.948 | 0.950 | 0.948 | 0.949 | 0.946 | 0.937 | 0.945 | 0.916 |
| 1  | 0.952 | 0.944 | 0.949 | 0.953 | 0.946 | 0.952 | 0.948 | 0.928 | 0.940 | 0.906 |
| 2  | 0.924 | 0.917 | 0.919 | 0.913 | 0.921 | 0.904 | 0.927 | 0.922 | 0.923 | 0.904 |
| 3  | 0.909 | 0.879 | 0.912 | 0.886 | 0.902 | 0.881 | 0.911 | 0.906 | 0.903 | 0.890 |
| 4  | 0.945 | 0.939 | 0.933 | 0.936 | 0.935 | 0.934 | 0.933 | 0.926 | 0.933 | 0.911 |
| 5  | 0.941 | 0.938 | 0.933 | 0.934 | 0.934 | 0.930 | 0.934 | 0.925 | 0.926 | 0.908 |
| 6  | 0.947 | 0.948 | 0.944 | 0.949 | 0.945 | 0.947 | 0.942 | 0.938 | 0.935 | 0.910 |
| 7  | 0.947 | 0.946 | 0.942 | 0.951 | 0.944 | 0.947 | 0.946 | 0.939 | 0.940 | 0.911 |
| 8  | 0.929 | 0.923 | 0.927 | 0.921 | 0.925 | 0.913 | 0.933 | 0.929 | 0.927 | 0.904 |
| 9  | 0.913 | 0.902 | 0.922 | 0.904 | 0.914 | 0.895 | 0.919 | 0.914 | 0.919 | 0.898 |
| 10 | 0.940 | 0.939 | 0.934 | 0.939 | 0.936 | 0.930 | 0.933 | 0.924 | 0.928 | 0.909 |
| 11 | 0.937 | 0.939 | 0.927 | 0.937 | 0.937 | 0.926 | 0.935 | 0.930 | 0.925 | 0.911 |
| 12 | 0.940 | 0.939 | 0.936 | 0.943 | 0.937 | 0.936 | 0.933 | 0.924 | 0.927 | 0.907 |
| 13 | 0.943 | 0.937 | 0.919 | 0.944 | 0.938 | 0.943 | 0.929 | 0.936 | 0.933 | 0.907 |
| 14 | 0.934 | 0.932 | 0.933 | 0.931 | 0.928 | 0.925 | 0.933 | 0.923 | 0.926 | 0.904 |
| 15 | 0.920 | 0.916 | 0.927 | 0.920 | 0.924 | 0.906 | 0.930 | 0.920 | 0.923 | 0.904 |
| 16 | 0.937 | 0.942 | 0.932 | 0.938 | 0.936 | 0.932 | 0.928 | 0.920 | 0.925 | 0.901 |
| 17 | 0.938 | 0.935 | 0.932 | 0.939 | 0.933 | 0.925 | 0.933 | 0.924 | 0.928 | 0.911 |
| 18 | 0.933 | 0.927 | 0.932 | 0.933 | 0.928 | 0.928 | 0.914 | 0.908 | 0.916 | 0.889 |
| 19 | 0.935 | 0.926 | 0.917 | 0.940 | 0.936 | 0.930 | 0.921 | 0.918 | 0.923 | 0.901 |
| 20 | 0.939 | 0.931 | 0.935 | 0.932 | 0.931 | 0.930 | 0.931 | 0.922 | 0.926 | 0.897 |
| 21 | 0.929 | 0.926 | 0.929 | 0.927 | 0.924 | 0.917 | 0.932 | 0.923 | 0.926 | 0.904 |
| 22 | 0.934 | 0.938 | 0.930 | 0.935 | 0.932 | 0.929 | 0.921 | 0.915 | 0.921 | 0.885 |
| 23 | 0.936 | 0.936 | 0.916 | 0.934 | 0.930 | 0.921 | 0.931 | 0.920 | 0.925 | 0.899 |

Mean: 0.9273, STD: 0.0144

Table 11: Clean accuracy of models trained on CIFAR-10-B0-20. The horizontal numbers represent the numbering of different network structures and the vertical numbers represent the numbering of different training hyperparameters.

|    | 0 | 1 | 2 | 3 | 4 | 5 | 6 | 7 | 8 | 9 |
|----|-------|-------|-------|-------|-------|-------|-------|-------|-------|-------|
| 0  | 0.948 | 0.951 | 0.948 | 0.951 | 0.947 | 0.953 | 0.950 | 0.939 | 0.944 | 0.913 |
| 1  | 0.951 | 0.949 | 0.949 | 0.952 | 0.949 | 0.950 | 0.949 | 0.931 | 0.942 | 0.910 |
| 2  | 0.926 | 0.913 | 0.925 | 0.909 | 0.921 | 0.907 | 0.928 | 0.926 | 0.926 | 0.906 |
| 3  | 0.905 | 0.881 | 0.904 | 0.887 | 0.895 | 0.882 | 0.909 | 0.900 | 0.910 | 0.892 |
| 4  | 0.941 | 0.938 | 0.935 | 0.935 | 0.935 | 0.929 | 0.933 | 0.927 | 0.928 | 0.913 |
| 5  | 0.939 | 0.934 | 0.927 | 0.929 | 0.936 | 0.929 | 0.936 | 0.927 | 0.926 | 0.914 |
| 6  | 0.945 | 0.945 | 0.944 | 0.947 | 0.941 | 0.946 | 0.945 | 0.935 | 0.937 | 0.910 |
| 7  | 0.949 | 0.944 | 0.944 | 0.949 | 0.945 | 0.948 | 0.949 | 0.938 | 0.937 | 0.913 |
| 8  | 0.931 | 0.924 | 0.930 | 0.920 | 0.925 | 0.914 | 0.930 | 0.924 | 0.926 | 0.904 |
| 9  | 0.914 | 0.904 | 0.915 | 0.910 | 0.917 | 0.895 | 0.919 | 0.915 | 0.912 | 0.900 |
| 10 | 0.941 | 0.941 | 0.931 | 0.938 | 0.933 | 0.931 | 0.935 | 0.924 | 0.929 | 0.909 |
| 11 | 0.939 | 0.933 | 0.924 | 0.936 | 0.933 | 0.928 | 0.937 | 0.927 | 0.928 | 0.907 |
| 12 | 0.940 | 0.937 | 0.932 | 0.942 | 0.935 | 0.939 | 0.929 | 0.924 | 0.929 | 0.905 |
| 13 | 0.942 | 0.938 | 0.935 | 0.943 | 0.940 | 0.944 | 0.935 | 0.933 | 0.933 | 0.909 |
| 14 | 0.932 | 0.931 | 0.930 | 0.926 | 0.930 | 0.927 | 0.931 | 0.927 | 0.925 | 0.905 |
| 15 | 0.920 | 0.921 | 0.926 | 0.915 | 0.917 | 0.909 | 0.926 | 0.919 | 0.921 | 0.905 |
| 16 | 0.937 | 0.938 | 0.929 | 0.934 | 0.935 | 0.933 | 0.928 | 0.917 | 0.924 | 0.899 |
| 17 | 0.941 | 0.937 | 0.926 | 0.935 | 0.931 | 0.926 | 0.935 | 0.924 | 0.929 | 0.911 |
| 18 | 0.934 | 0.930 | 0.928 | 0.937 | 0.929 | 0.930 | 0.917 | 0.909 | 0.918 | 0.891 |
| 19 | 0.935 | 0.918 | 0.919 | 0.939 | 0.929 | 0.930 | 0.921 | 0.922 | 0.922 | 0.902 |
| 20 | 0.936 | 0.934 | 0.934 | 0.927 | 0.930 | 0.930 | 0.931 | 0.920 | 0.922 | 0.898 |
| 21 | 0.933 | 0.927 | 0.929 | 0.926 | 0.926 | 0.912 | 0.930 | 0.924 | 0.927 | 0.901 |
| 22 | 0.935 | 0.939 | 0.931 | 0.937 | 0.931 | 0.931 | 0.923 | 0.912 | 0.920 | 0.890 |
| 23 | 0.935 | 0.934 | 0.921 | 0.935 | 0.931 | 0.929 | 0.930 | 0.921 | 0.924 | 0.903 |

Mean: 0.9272, STD: 0.0141

# H  EXPERIMENTAL RESULTS ON CIFAR-10 WHEN $y' = 5$

Here we consider the effectiveness of ETI when the attack target $y'$ is 5. We perform the exact same steps as $y' = 0$, and the experimental results are shown in Figure 11. It can be seen that it is more difficult to attack a target of 5 on CIFAR-10 than to attack a target of 0. The overall poisoning ratios required to achieve 90% attack success rates increase. Among these methods, ETI-generated poisoned samples are still the most efficient, with a ratio of about 0.069.

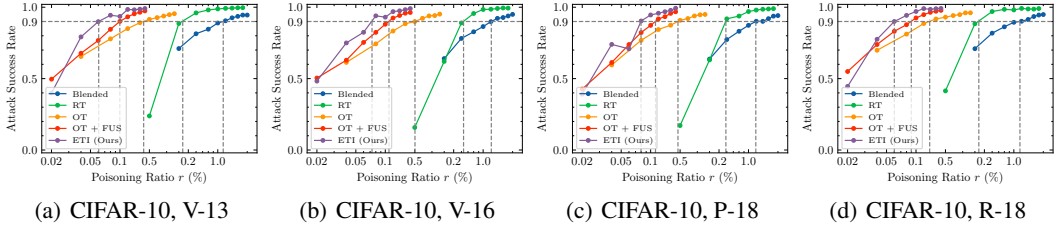

| (a) CIFAR-10, V-13 | (b) CIFAR-10, V-16 | (c) CIFAR-10, P-18 | (d) CIFAR-10, R-18 |
|---|---|---|---|

Figure 11: Attack success rates on CIFAR-10 when $y' = 5$.

Similarly, we verify that the individual consistency still holds when the target is 5, and the results are shown in Table 12 and Figure 12.

Table 12: Poisoning ratios $r$ (%) needed to achieve 90% attack success rates for the poisoned sample sets generated from 10 independent runs using OT + FUS when $y' = 5$.

|      | 0 | 1 | 2 | 3 | 4 | 5 | 6 | 7 | 8 | 9 |
|------|-------|-------|-------|-------|-------|-------|-------|-------|-------|-------|
| V-13 | 0.126 | 0.096 | 0.061 | 0.090 | 0.095 | 0.100 | 0.093 | 0.101 | 0.092 | 0.116 |
| V-16 | 0.147 | 0.105 | 0.073 | 0.089 | 0.103 | 0.129 | 0.104 | 0.108 | 0.104 | 0.118 |
| P-18 | 0.141 | 0.103 | 0.080 | 0.093 | 0.111 | 0.140 | 0.079 | 0.113 | 0.099 | 0.123 |
| R-18 | 0.101 | 0.086 | 0.060 | 0.070 | 0.074 | 0.096 | 0.071 | 0.097 | 0.089 | 0.100 |
| Mean | 0.129 | 0.098 | **0.069** | 0.086 | 0.096 | 0.116 | 0.087 | 0.105 | 0.096 | 0.114 |

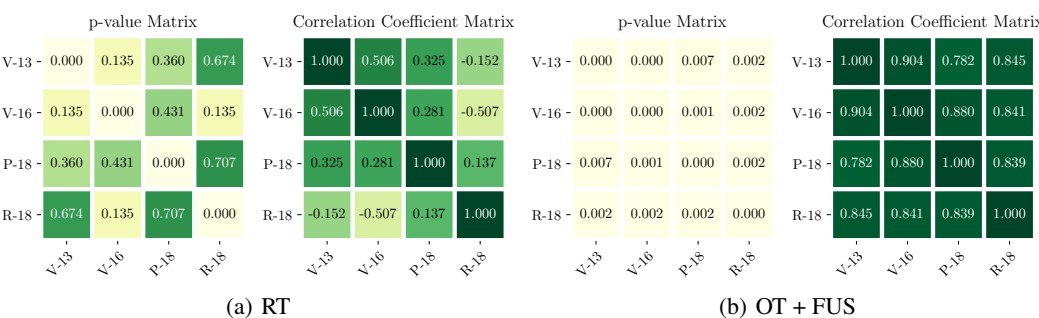

| (a) RT | (b) OT + FUS |
|---|---|

Figure 12: Pearson correlation analysis on both RT and OT + FUS on CIFAR-10 when $y' = 5$.

# I EXPERIMENTAL RESULTS ON CIFAR-10 WHEN $\epsilon = 10/255$ AND $\epsilon = 12/255$

The constraint of the trigger considered in this paper is $C(t) := \|v\|_\infty$. One parameter that may have a significant impact on the results is $\epsilon$. In the above experiments, the results given are in the case where $\epsilon = 8/255$, here we test two additional cases, i.e. $\epsilon = 10/255$ and $\epsilon = 12/255$, the results are as shown Table 13. It can be seen that just poisoning 10 images, ETI can achieve an attack success rate of 0.935 when $\epsilon = 10/255$ and 0.971 when $\epsilon = 12/255$.

Table 13: Attack success rates on CIFAR-10 with different $\epsilon$ when the poisoning ratio set to 0.02% (10/50,000).

|  | V-13 | V-16 | P-18 | R-18 | Mean |
|---|---|---|---|---|---|
| $\epsilon = 8/255$, OT + FUS | 0.705 | 0.685 | 0.681 | 0.806 | 0.719 |
| $\epsilon = 8/255$, ETI | 0.798 | 0.816 | 0.804 | 0.906 | 0.831 |
| $\epsilon = 10/255$, OT | 0.799 | 0.827 | 0.788 | 0.871 | 0.821 |
| $\epsilon = 10/255$, OT + FUS | 0.885 | 0.878 | 0.868 | 0.915 | 0.886 |
| $\epsilon = 10/255$, ETI | 0.895 | 0.942 | 0.951 | 0.952 | 0.935 |
| $\epsilon = 12/255$, OT | 0.903 | 0.906 | 0.903 | 0.936 | 0.912 |
| $\epsilon = 12/255$, OT + FUS | 0.948 | 0.952 | 0.953 | 0.969 | 0.956 |
| $\epsilon = 12/255$, ETI | 0.980 | 0.963 | 0.955 | 0.987 | 0.971 |

# J IN-DISTRIBUTION GENERALIZATION ANALYSIS

All the experiments above assume that the attacker has complete access to the clean dataset. Here, we assume that the attacker has access to only a portion of the dataset to verify the in-distribution generalization of the ETI-generated poisoned samples. We divide the training data of CIFAR-10 into two randomly disjoint subsets, CIFAR-10A and CIFAR-10B. Subsequently, we generate poisoned samples in these two subsets independently using the ETI method. Finally, we poison CIFAR-10A and CIFAR-10B with the generated poisoned data and test the performance of attacks. We define some symbols here. A2A means that the poisoned samples are generated on CIFAR-10A, and are used again to poison CIFAR-10A. A2B means that the poisoned samples are generated on CIFAR-10A, but are used to poison CIFAR-10B.

We first observe whether the individual consistency can exist across subsets. This is important because the attacker can only select individuals based on what he has on hand and expects the outstanding individual to still perform well on a different subset. Here, we conduct 10 independent runs with OT + FUS on the CIFAR-10A subset and calculate the correlation coefficients among different models, and the results are shown in Figure 13. It can be seen that the individual consistency is still maintained very well even across different subsets, which is a very good characteristic for attackers.

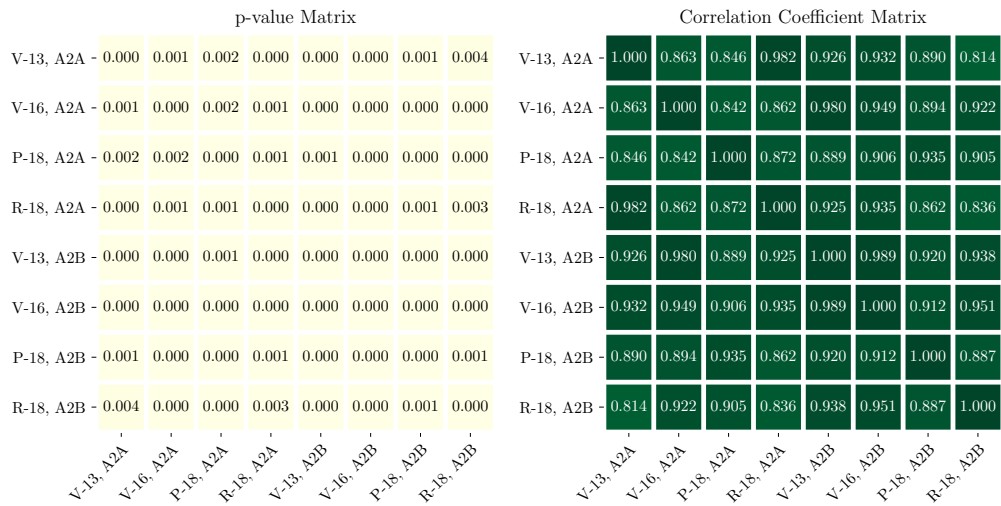

Figure 13: Pearson correlation analysis on OT + FUS on CIFAR-10 under cross-subset conditions.

Finally, we test the performance of the A2B attacks, as shown in Figure 14. We use the results of B2B attacks as comparisons. It can be seen that the poisoned samples generated by ETI have a fairly good in-distribution generalization: there is almost no difference between the performance of A2A attacks and B2B attacks.

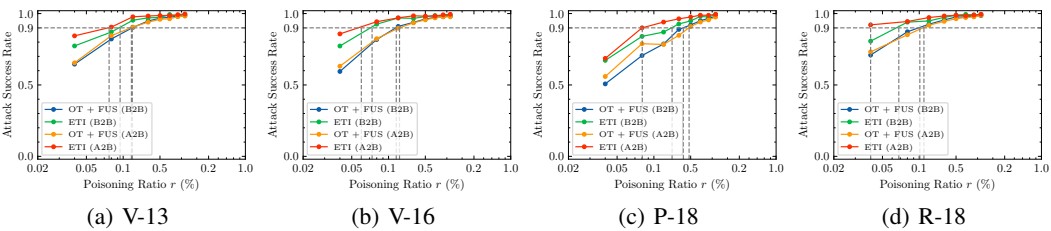

Figure 14: Attack success rates on CIFAR-10B. All curves (except ETI) are averaged over 10 independent runs.

