# OpenReview forum: "Efficient Trojan Injection: 90% Attack Success Rate Using 0.04% Poisoned Samples"
_ICLR.cc/2023/Conference — Submitted to ICLR 2023_

### Official Review · Reviewer_F4UP · 2022-10-24

**Confidence:** 3
**Clarity, Quality, Novelty And Reproducibility:** see Strength And Weakness.
**Correctness:** 3
**Technical Novelty And Significance:** 2
**Empirical Novelty And Significance:** 2
**Recommendation:** 5

**Strength And Weaknesses:**

**Strength**
1. It is always interesting to explore the capability limits of data poisoning, i.e., the minimum amount of data we can use to inject backdoor behavior into DNN.
2. The experimental results are promising. We can use only 20 poisoned images to successfully inject backdoor behavior (50K images on training data in total).


**Weakness**
1. The technical novelty in this paper is limited. The proposed method looks like a trivial combination of existing methods like universal adversarial patch and FUS.
2. I have concerns about the threat model of this paper.
- Is the threat model of this paper realistic? In this threat model, the adversary has full access to the training data, and use the knowledge from the whole data to select poisoned data.
Can authors show whether this method could generalize across in-distribution data?
- Can the authors verify whether the proposed method generalizes across in-distribution data? For example, we can split the training dataset into two parts. We can select the poisoned data from the first part using the proposed method, and verify the effectiveness on the second part.
3. FUS is an important technique in this paper. To make the paper more self-containing, can the authors add descriptions of FUS with more details in Related Work. Otherwise, it is hard for readers to understand this method.
4. In my opinion, the selection of the target class has a significant impact on the poisoning performance. This paper only considers the cases when class 0 is the target class. This is incomprehensive and makes the results unconvincing because the adversary might choose any class as the target class (the target class should first satisfy the attack aim for the adversary)


**Summary Of The Paper:**

This paper studies the capability limits of data poisoning, i.e., the minimum amount of data required to successfully inject backdoor behavior into DNN. In particular, the authors craft a universal adversarial patch as the trigger, use FUS to select samples, and exploit individual consistency to stabilize the performance. Finally, they construct a poisoned CIFAR-10 training data (50K in total), where only 20 images are poisoned and the model still learns the backdoor behaviors.

**Summary Of The Review:**

Although the research topic of this paper is interesting, the proposed method lacks technical contributions. If the authors could have a much closer look at the proposed method and provide more insights, I will reconsider my score.

---

> ### Author Response · Authors · 2022-11-14
> **Response to Reviewer F4UP**
>
> Thanks for your throughful suggestions!
>
> > Novelty
>
> Please see our general response.
>
> > In-distribution generalization
>
> We think this is a very worthwhile issue to explore. Intuitively, we believe that the Optimized Trigger (OT) and Filtering-and-Updating Strategy (FUS) can do this, because Universal Adversarial Perturbation (UAP) itself has some transferability, and FUS involves only a portion of the entire dataset when searching. But we do not know whether the individual consistency is maintained across subsets. So we verify these through experiments. We randomly divide the CIFAR-10 training set into two disjoint subsets A and B. Then we use ETI to generate the poisoned samples on A, and poison B with these samples to construct the backdoor attacks (A2B); Similarly, we also construct the B2B attacks. The poisoning ratios $r$ (%) needed to achieve 90% attack success rates are as follows.
>
> |                          | V-13 | V-16|P-18|R-18|Mean|
> | :-----| :----: | :----: |:----: |:----: |:----: |
> |A2B, OT+FUS   |0.117 |0.123|0.194|0.109|0.136|
> |A2B, ETI (Ours) |0.077|0.06|0.08|0.04|0.064|
> |B2B, OT+FUS   |0.119 |0.116|0.174|0.101|0.128|
> |B2B, ETI (Ours) |0.094|0.073|0.141|0.068|0.094|
>
> It can be seen that A2B, ETI is even better than B2B, ETI (we think this is due to randomness). These results demonstrate that ETI has good in-distribution generalization. For more detailed experimental setup and results, please see our revised manuscript, Appendix J.
>
> > Details of FUS
>
> We have added descriptions about FUS on Related Work. Also, we have given the detailed procedure of this algorithm in Appendix C.
>
> > Different targets of backdoor attacks
>
> We set the attack target  for CIFAR-10 as class 5 and test the effectiveness of ETI. The poisoning ratios $r$ (%) needed to achieve 90% attack success rates are as follows.
>
> || V-13 | V-16|P-18|R-18|Mean|
> | :-----| :----: | :----: |:----: |:----: |:----: |
> |Blended|1.137|1.212|1.188|1.154|1.173|
> |Random Trigger (RT)|0.439|0.631|0.588|0.439|0.524|
> |OT|0.175|0.201|0.19|0.138|0.176|
> |OT+FUS|0.1|0.109|0.112|0.09|0.103|
> |ETI (Ours)|0.061|0.073|0.08|0.06|0.069|
>
> It can be seen that ETI is effective when the attack target is 5. For more detailed experimental results, please see our revised manuscript, Appendix H.

---

> > ### Comment · Reviewer_F4UP · 2022-12-11
> > **Thank you for reply**
> >
> > The proposed method behaves well in the in-distribution generalization according to the experiments. I believe this is a good strategy for conducting efficient data poisoning. However, as an academic paper, this work still lacks novelty. The combination of optimized triggers and importance sample selection is trivial. Although there might be some thoughts on randomness, unfortunately, this tiny novelty is unable to contribute enough to this community and make this paper accepted by top conferences like ICLR. Thus, I will keep my score.

---

### Official Review · Reviewer_MNku · 2022-10-25

**Confidence:** 4
**Correctness:** 2
**Technical Novelty And Significance:** 2
**Empirical Novelty And Significance:** 2
**Recommendation:** 5

**Clarity, Quality, Novelty And Reproducibility:**

Clarity:

(-) The "inherent flaws" of neural networks were mentioned multiple times throughout the paper and also served as the motivation of the proposed algorithm. However, I didn't find any explanation for these flaws except for the references. It would be helpful if details or intuition of these inherent flaws could be added.

(-) Similarly, most previous works are not well-introduced but stopped at generic terms like "some factors" (what factors?), "their proposed strategy" (what strategy?).

(-) The main baseline _RT_ is not introduced. Does it stand for "Random Trigger"? How did the random perturbations generated?

A few minor questions/issues:
* Equation 1: _D_b_ is not defined.
* Figure 4: What is the _difference_? Is it the _difference between the attack success rates_?
* _ET_ was not defined until the end of Section 3.2 but was used throughout Section 3.1.

Quality:

(+) The proposed method can attack a neural network with a very small amount of data and the backdoored datasets are effective for models with different network architectures and trained with different optimizers.

(-) The metric used for evaluation ("attack success rate") is not formally defined or introduced. It is also unclear what the test data is.

(-) The clean accuracy (when the trigger is not added to the examples) is not reported. In this case, it is hard to conclude whether the generated trigger is effective or not.

(-) All experiments used the same target label which is category 0. I would suggest to add results of using another attack target _y'_ to validate the effectiveness of the proposed ETI attack.

Novelty:

(-) The novelty of this work is unclear. Previous work related to the first contribution (Section 3.1) is only vaguely mentioned and thus it is unclear how the proposed method is different from existing ones. The claimed improvement over FUS (Xia et al., 2022) in Section 3.2 also seem to be trivial. It is merely saving the best instead of the last result. If this paper is supposed to be an emprical study on the smallest number of examples needed for backdoor attacks, more ablation studies and analysis of factors that affect this number should be provided.

Reproducibility:

(-) Code or backdoored datasets are not released.

**Strength And Weaknesses:**

Strength:
- The problem studied in this paper, data efficiency of backdoor attacks, is important and of the interest of the ICLR community.
- The claimed result is significant in terms of how small the set of poisoned data can be to make a trigger effective for the poisoned model.

Weaknesses:
- The novelty of this paper is unclear. There could be more discussion about the background and existing work to show how this work is different from the existing ones.
- The writing of this paper is lacking as a conference submission. The authors need to carefully check the definitions to make sure all notations and terms are defined when they first appear in the paper. The language used in this paper can also be more formal.
- The claimed result lack supports to make it convincing. Direct comparison to existing work, experiment details and experiments with different targeted labels should at least be added.

**Summary Of The Paper:**

This paper studied the data efficiency of backdoor attacks and found that one only needs to add an adversarially optimized trigger to 0.04% and 0.06% of training examples to have a high attack accuracy on CIFAR-10 and CIFAR-100 respectively. These numbers are much smaller than the ones reported in previous work.

**Summary Of The Review:**

I found the problem studied and the result presented in this paper interesting. However, a lot of information, including a comparison to existing work and experiment details, is currently missing, and the effectiveness of the proposed method is not well-supported by enough experiments. Therefore, I recommend rejecting this manuscript.

Given the authors' response during the discussion, I would like to raise my score from 3 to 5.

---

> ### Author Response · Authors · 2022-11-14
> **Response to Reviewer MNKu**
>
> Thanks for your comments.
>
> > Novelty
>
> Please see our general response.
>
> > Inherent flaws
>
> Inherent flaws here refer to the perturbations that can be found on a **clean** model to deceive the network, i.e., universal adversarial perturbations. In the revised manuscript, we explicitly explain the meaning of inherent flaws in section 2.2.
>
> > "some factors", "their proposed strategy"
>
> We explain each of these words in the follow-up sections. "some factors" is explained in Sec. 2.1, para. 2. "their proposed strategy" is the Filtering-and-Updateing Strategy (FUS). In the revised version, we have partially rewritten the Introduction section to make clear how our work relates to and differs from previous work.
>
> > Baseline RT
>
> A random trigger is a randomly generated perturbation of the same size as the input image, where its $l_{\infty}$-norm is bounded between [$-\epsilon$, $\epsilon$].
>
> > $D_b$
>
> We actually define $D_b$, at the beginning of Section 2.1: "As shown in Figure 1, given a benign training set $D_b$, the attacker builds the mixed training set $D_m$ in three steps.".
>
> > Figure 4: What is the difference?
>
> You are right, it means the difference between the attack success rates. We have added it in the revised version. Thanks.
>
> > ET was not defined until the end of Section 3.2
>
> Sorry for that. We have checked and revised the manuscript to avoid similar issues.
>
> > Attack success rate
>
> This is a very basic concept in backdoor attacks. It refers to the proportion of the poisoned test set that is classified as the attack target $y'$ by the trained infected model. The poisoned test set is constructed based on the clean test set. For example, we test the attack success rate on CIFAR-10-B0-20 by constructing the poisoned test set using the clean validation set of CIFAR-10 with the same poisoning method $F(x,t)$ as the backdoor injection phase.
>
> > Clean accuracy
>
> Thank you for your suggestion. We have added a comparison of clean accuracy on CIFAR-10 and CIFAR-10-B0-20 in Appendix G. The ETI-generated poisoned samples have almost no effect on the clean accuracy.
>
> > Different targets of backdoor attacks
>
> In the revised manuscript, we set the attack target for CIFAR-10 as class 5 and test the effectiveness of ETI. Please see our response to Reviewer F4UP and Appendix H.
>
> > Reproducibility
>
> We will upload the code and datasets to the supplementary materials before the end of the rebuttal phase.

---

> > ### Comment · Reviewer_MNku · 2022-12-12
> > **Thank you for the authors' response**
> >
> > I appreciate the authors' direct response to my concerns and the improvement they made in the updated manuscript over the original version.
> >
> > For the definitions and clarifications I asked for, I'm not asking them for myself but from a reader's perspective. I would like to thank the authors for explaining them here to me in this response, but it would be helpful if the explanations could also be added to the manuscript.
> >
> > Given the improved manuscript and the authors' discussion with other reviewers, I have raised my score from 3 to 5. I'm still leaning toward the rejection side because the novelty and technical contributions of this paper are still marginal, at least in the current form of presentation.

---

### Official Review · Reviewer_rDHq · 2022-10-25

**Confidence:** 4
**Correctness:** 4
**Technical Novelty And Significance:** 2
**Empirical Novelty And Significance:** 2
**Recommendation:** 3

**Clarity, Quality, Novelty And Reproducibility:**

Generally, the writing is relatively clear and in a logical structure. However, the novelty is limited as mentioned above.

**Strength And Weaknesses:**

Strengths:
- The paper demonstrated good experimental results and showed clear comparisons of the performance of the proposed methods and the baseline approach.
- Detailed descriptions of the background of the backdoor attacks are provided to make the work of the paper more understandable.
- The paper is well written and the empirical results are reasonable.

Weaknesses:
The main concern of the paper is about the novelty and research contribution. Moreover, the involved techniques to improve backdoor attacks added extra assumptions of the knowledge of the attacker. For example,
- The process of generating the trigger is based on the idea of “universal attack”. However, in other works about backdoor attack, the attacker has the flexibility to choose customized triggers (i.e., with arbitrary choices of shape, color) without having access to the targeted model.
- During the sample selecting process, this paper directly applies Xia et al (2022) to perturb the “forgettable” samples. However, this process may involve a filtering process, which is also potential to highly demand the knowledge of the attacker.

If the attacker has sufficient capacity to achieve the two requirements above, it is not very surprising that an attacker can greatly improve the effectiveness of backdoor attacks.


**Summary Of The Paper:**

The paper investigates three separated techniques to improve the successful rate and perturbation efficiency of backdoor attacks. Based on the proposed attacking strategy, the authors constructed two backdoored datasets based on CIFAR-10 and CIFAR-100, in which 0.04% and 0.06% of the data are poisoned. Both of them have achieved attack success rates higher than 90% on models with different architectures and hyperparameters.

**Summary Of The Review:**

The paper introduces a new pipeline for conducting backdoor attacks which empirically showed a good performance in reducing the number of poisoned samples required. However, it may lack potential impact to the community, because it only combines existing techniques to improve the performance, and also only rely on strong assumptions on the knowledge of the attacker.

---

> ### Author Response · Authors · 2022-11-14
> **Response to Reviewer rDHq**
>
> Thank you for your comments.
>
> > Novelty
>
> Please see our general response.
>
> > Customized triggers vs. optimized triggers
>
> Our view on this point is that an attacker can choose to use an arbitrary trigger, but needs to construct more poisoned samples to complete the attack, which increases the possibility of the attack being detected; or he can improve the poisoning efficiency by optimizing the trigger. The two are not in conflict.
>
> It is also important to note that here we have tried to limit the attacker's capabilities. We assume that the attacker uses only one model in constructing the poisoned data - VGG13 - while the user can use a completely different model structure and hyperparameters.
>
> >  The sample selecting process also potentially highly demands the knowledge of the attacker.
>
> We agree with you that FUS does require the attacker to be able to control the **entire** dataset. In the revised version, we try to assume that the attacker has only part of the data and find that ETI can also generate efficient poisoned samples. Please see our response to Reviewer F4UP (# in-distribution generalization part) and Appendix J.
>
> ------------
>
> **A discussion about the attacker's knowledge and the poisoning efficiency**
>
> We believe that it is worthwhile to study the poisoning efficiency even if the current assumptions require more knowledge of the attacker. The work we do is actually to explore the limit of the capability. We subsequently reduce the attacker capability (from controlling the whole dataset to a partial dataset) and the poisoning efficiency can still be improved. Perhaps the attacker is able to improve the efficiency of the attack without or with weaker assumptions? This is what we would like to do in the future.

---

> > ### Comment · Reviewer_rDHq · 2022-12-11
> > **Response to the rebuttal**
> >
> > Thanks for your response and we decide to keep the original score as:
> > 1. The combination from FUS + optimized trigger is limited.
> > 2. The problem about the **randomness** in backdoor is interesting and we believe that it worths deeper studying. However, this paper lacks sufficient discussions about this point.

---

### Official Review · Reviewer_eNSJ · 2022-10-28

**Confidence:** 4
**Clarity, Quality, Novelty And Reproducibility:** See above.
**Correctness:** 3
**Technical Novelty And Significance:** 2
**Empirical Novelty And Significance:** 3
**Recommendation:** 6

**Strength And Weaknesses:**

This paper is very well written and to-the-point.  Attacking efficiency is an important problem, which is largely ignored by previous literature. Although the proposed ideas are mostly based on existing ideas (see below), the empirical study seems thorough and the paper provides important insights. I do appreciate this paper more than many other papers on backdoor attacks.

Regarding novelty, the second idea (ISS) is mainly based on (Xia et al. 2022). Meanwhile, the first idea is a bit incremental considering adversarial perturbation for trojan attack has been explored. The relationship with the universal adversarial perturbation (Moosavi-Dezfooli et al. 2017) should also be discussed.

* Pang, Ren, Hua Shen, Xinyang Zhang, Shouling Ji, Yevgeniy Vorobeychik, Xiapu Luo, Alex Liu, and Ting Wang. "A tale of evil twins: Adversarial inputs versus poisoned models." In Proceedings of the 2020 ACM SIGSAC Conference on Computer and Communications Security, pp. 85-99. 2020.

However, this does not affect the novelty of the empirical insights.

Aside from the poisoning rate, I consider the third observation (consistency) very interesting. Yet, at current version it is under-explored. I would hope to see more discussions of how the observation implies in terms of model prediction consistency. Some more analysis could be very useful. For example, any statistics as to the proportions of samples that are consistently misclassified across different runs? How are the successful triggered test samples related to the poisoning samples?

Questions to be addressed.
1) in Fig 6, please explain how the correlation is computed (sample-wise?)
2) adversarial training is known to be a challenge for poisoning. How would the attack perform when the user employs adversarial training?
3) for the idea #2, what if the training trial to choose samples are different from the training in terms of setting, e.g., different optimizers/learning rates/standard vs adversarial?
4) how would the created poison samples perform if the user chooses a different model architecture?

Some limited ablation study to address these questions could be helpful.




**Summary Of The Paper:**

This is a practical paper providing empirical insights on how to better backdoor attack a DNN model via poisoning samples (without interfering with the training process). Three observations/ideas are provided: 1) using shared adversarial perturbation as the trigger; 2) using easiness of being misclassified through a trial training to select poisoning samples; 3) the poisoning sample sets produced by 1) and 2) tend to provide consistent poisoning effect.

**Summary Of The Review:**

Incremental ideas. Novel empirical insights. Important problem and relatively thorough study (with limitations). Paper is well written.

---

> ### Author Response · Authors · 2022-11-14
> **Response to Reviewer eNSJ**
>
> Thanks for the positive comments to our work.
>
> >  The relationship with the universal adversarial perturbation (Moosavi-Dezfooli et al. 2017) should also be discussed.
>
> In the revised manuscript, we have discussed it in Sec 2.2.
>
> > More analysis about the individual consistency
>
> We also find the consistency is interesting. We have tried to do more analysis.
>
> **1. Counting the proportion of samples that are consistently misclassified across different runs.**
>
> We count the proportion of samples that are successfully attacked in all 10 runs, unsuccessfully attacked in all 10 runs, and successfully attacked in only 1 run, as shown as follows.
>
> ||V-13|V-16|P-18|R-18|
> |:-------------|:-------------:|:-------------:|:-------------:|:-------------:|
> | All successful      | 58.8% | 54.4% | 54.3% | 70.2% |
> | All fail                  | 3%      | 4.9% | 3.6% | 1.9%|
> | Successful 1 run | 2.2%   | 3.2% | 2.5%| 1.7% |
> | Other situations  | 36%    | 37.5% | 39.6%| 26.2%|
>
> About 50% to 70% of the test poisoned samples are successfully attacked in all runs. We had hoped that the percentage of samples that were successfully attacked just 1 run would be high, but this did not occur.
>
> **2. Observation of the consistency across subsets**
>
> We observe the individual consistency when studying the in-distribution generalization of ETI. The specific assumption is as follows: the attacker has only one partial dataset on which he can only construct poisoned samples. Then, when he tries to poison these constructed samples on another partial dataset that has not been seen before but belongs to the same distribution, is the efficiency of the samples still guaranteed? Naturally, we wonder if the individual consistent performance can also exist across different subsets and different models. We experiment and find that the consistency can be maintained very well. The details can be seen in our response to Reviewer F4UP (# in-distribution generalization part) and Appendix J.
>
> > In Fig 6, please explain how the correlation is computed (sample-wise?)
>
> We use each row in Table 2 to characterize the performance of the model under different runs, forming a vector. After that, we calculate the correlation coefficients between the different models (vectors) two by two and obtain Figure 6.
>
> > How would the attack perform when the user employs adversarial training?
>
> We test the attack performance of the constructed poisoned samples under $l_{\infty}$, $\epsilon=4/255$ adversarial training. The results show that backdoor injection is almost unsuccessful under adversarial training. We then try to use the adversarial pre-trained model at the time of constructing the trigger and find that it is almost impossible to find a suitable UAP. How to efficiently inject backdoors under adversarial training is indeed a worthwhile research problem.
>
> > Different optimizers/learning rates/standard vs adversarial? Different model architectures.
>
> Our answer to the last question illustrates that the performance of the poisoned samples under standard and adversarial training is quite different. For the other conditions, we are able to experimentally demonstrate that the ETI-generated samples are still efficient. In fact, we do so for all the experimental results shown in the manuscript. In generating the poisoned samples using ETI, we assume that the attacker uses **only** the VGG-13 model, the optimizer is chosen to be Adam, and the initial learning rate is set to 0.001. For testing, we train the infected models on the **same** poisoned sample set with different DNN architectures, different optimizers, different batch sizes, and different initial learning rates (see Appendix A).

---

### Public Comment · ~Minzhou_Pan1 · 2022-11-09
**What is the difference between this method and the Narcissus clean-label backdoor attack?**

I appreciate your submission. After reading the paper, I observed a striking resemblance between your formulation with the one in [1].
In particular, the trigger-generating backdoor formulation in [1] is :
$$
\underset{\delta \in \Delta}{\min } \sum_{(x, y) \in D} \mathcal{L}\left(f_{\theta}(x+\delta), y\right)
$$
whereas in your paper, you formulate the trigger as follows:
$$
\min\_{C(t) \leq \epsilon} \sum_{(x, y) \in \mathcal{D}_b} L\left(f_\theta(x+t)), y^{\prime}\right)
$$
Both seem to utilize a pre-trained model to generate noise that minimizes loss on the target label. Could you explain how theoretically, your contribution differs from [1]?

I hope to get some clarification. Thank you for your time!

[1] Zeng, Yi, et al. "NARCISSUS: A Practical Clean-Label Backdoor Attack with Limited Information." arXiv preprint arXiv:2204.05255 (2022).

---

> ### Author Response · Authors · 2022-11-10
> **Reply to "What is the difference between this method and the Narcissus clean-label backdoor attack?"**
>
> Thank you for your comments.
> Indeed, we have shown that this trigger generation is an existing technique, as in the paper we cited [1][2].
>
> [1] Backdoor Embedding in Convolutional Neural Network Models via Invisible Perturbation.
>
> [2] Clean-Label Backdoor Attacks on Video Recognition Models.
>
> What we are trying to do here is to systematically improve the poisoning efficiency, including trigger design, sample selection, and focusing on randomness, to see how many poisoned samples an attacker can use to complete a backdoor attack at this point.

---

> > ### Public Comment · ~Minzhou_Pan1 · 2022-11-10
> > **Reply to your ''Reply''**
> >
> > Thanks for your response.
> >
> > *
> > Firstly, I think the authors should correctly include this work in the reference.
> >
> > *
> > Additionally, as evaluated in the original work of Zeng et al., they have already achieved an ASR above 97% by using a poison ratio of less than 0.05%. In particular, they can achieve a 99.89% by manipulating only **three** images on the PubFig dataset. On CIFAR-10, the same setting was evaluated in this work. They can reliably achieve 97.36% ASR in a clean-label way by manipulating 25 images (*5 more images, then earning a perfect attack efficacy*). I highly recommend the authors incorporate this work into the comparison group.
> >
> > *
> > Last, I find the authors slightly misunderstand the two references discussed here (i.e., [1,2]). The two cited works **do not strictly share the same trigger generation formulation** as Zeng et al. and what has been discussed in this work (which we find are **highly similar and not correctly cited**).
> > [1,2] adopts a deepfool-like (or the classic universal adversarial perturbation generating approach) approach as the backbone to synthesize the trigger. The trigger is gradually updated by taking the gradient one by one on each sample and updating the trigger sample-wisely. However, in Zeng et al. and this work, the two formulations aim to synthesize $\delta$ directly through the dataset, i.e., taking the gradient of a population of data and taking the average (instead of obtaining the gradient in a sample-wise way).
> >
> > ***
> > **Example:**
> >    Specifically, assuming $P_{p, \xi}(\cdot)$ is the norm projection of the gradient regarding $p$-norm with a constraint $\xi$ (non-linear), and $\delta$ is the synthesized trigger. Given an initialized noise, $\delta_0$, and a batch of data with size $N$:
> >
> >    * **In [1,2]**, the individual gradient associated with the first data is evaluated at the initial perturbation $\delta_0$:
> >
> >      $G_1 = \nabla_{\delta} L(x_1+\delta, \theta) |_{\delta=\delta_0}.$
> >
> >      The resulting perturbation is
> >
> >      $\delta_1 = P_{p, \xi}(\delta_0 + G_1).$
> >
> >      Note that the gradient for the second data point is computed at $\delta_1$ instead of $\delta_0$:
> >
> >      $G_2 = \nabla_{\delta} L(x_2+\delta, \theta) |_{\delta=\delta_1}.$
> >
> >      $\delta_2 = P_{p, \xi}(\delta_1 + G_2).$
> >
> >      $\vdots$
> >
> >      In [1,2], or the deepfool-like approach, the updated trigger of each data point is calculated sequentially; hence, the gradient for each data point in the batch is evaluated at different perturbations. The resulting perturbation is also highly dependent on the order of loaded data. Whatsmore, the norm ball projection is conducted $N-1$ times, resulting in the loss of large values regarding individual samples during the updates.
> >    * **In Zeng et al. and this work**, the unique and similar design has resulted in a much similar design of obtaining the trigger:
> >
> >      $G = \sum_{i=1}^N \nabla_{\delta} L(x_i + \delta, \theta) |_{ \delta=\delta_0},$
> >
> >      where all the gradients are computed at $\delta=\delta_0$.
> >
> >      $\delta = P_{p, \xi}(\delta_0 + G),$
> >
> >      where norm ball projection is only adopted once.

---

> > > ### Author Response · Authors · 2022-11-10
> > > **Reply**
> > >
> > > Thank you for your suggestion and discussion.
> > >
> > > + We will include Zeng et al. in the next revision.
> > >
> > > + In the work of Zeng et al., 97.36% ASR is achieved using 25 samples on CIFAR-10, with the linf constraint set to 16/255. However, we use 8/255 in this paper. We will add the 16/255 case as a comparison in the next revision.
> > >
> > > + We agree that the work of Zeng et al. and [1, 2] are not **exactly** the same. But as far as the objective function is concerned, it is very similar. The difference may lie in the specific optimization algorithms; [1, 2] used stochastic projected gradient descent, while Zeng et al. used projected gradient descent. In fact, what we use is mini-batch stochastic projected gradient descent. Different algorithms may have some impact on the efficiency of the trigger.
> > >
> > > Thanks.

---

> > > > ### Comment · Reviewer_eNSJ · 2022-12-03
> > > > **I do not see why this arxiv paper should be cited**
> > > >
> > > > Not that this is relevant to the ongoing reviewing of this manuscript, but I found the request to cite an unpublished and recent arxiv paper unjustifiable. The referred paper has not been published in any peer-reviewed conference/journal and was submitted to arxiv in April 2022, only 5 months before the manuscript was submitted to ICLR. By any standard, I do not think the authors have an obligation to cite this arxiv paper and to explain the difference.
> > > >
> > > > The authors are already stressful enough to provide clarifications and additional experiments. We should all be considerate and not bother them on issues unrelated to the reviewing. (If this arxiv paper has been published officially or has been available online for sufficient amount of time, I would think this post is relevant to the reviewing process.)

---

> > > > ### Comment · Reviewer_F4UP · 2022-12-06
> > > > **The difference between studies [1, 2] and Zeng et al. with respect to the trigger generation**
> > > >
> > > > Dear Minzhou Pan,
> > > >
> > > > After reading your discussion, I think previous studies [1, 2] and Zeng et al. attempt to solve the same optimization problem using two slightly different methods. Specifically, previous studies [1, 2] use a multi-step iterative approach with estimated gradients based on batch data. Meanwhile, Zeng et al also use a multi-step method with an "accurate" gradient on the whole data (see Algorithm 1 in Zeng et al). Do I understand it correctly?
> > > >
> > > > Besides, I have two more questions:
> > > >
> > > > 1. *"The resulting perturbation is also highly dependent on the order of loaded data."*
> > > >
> > > > As we all know, the estimated gradient, which is calculated on randomly sampled batch data, achieves great success in many applications, even including SGD. It is interesting to think about the influence of the data order. Is there any theoretical or empirical evidence to show that the generation results fluctuate severely in random sampling (especially after several iterations)?
> > > >
> > > > 2. *"Whatsmore, the norm ball projection is conducted  times, resulting in the loss of large values regarding individual samples during the updates."*
> > > >
> > > > Is there any theoretical or empirical evidence to show the effects caused by the loss of large values?
> > > > Besides, according to Algorithm 1 in Zeng et al., the norm ball projection is conducted N times (Line 4), where N is the number of iterations. Thus, do Zeng et al. also suffer from this problem?
> > > >
> > > > Since I cannot see the difference between previous studies [1, 2] and Zeng. et al, I don't see the necessity to cite this arxiv paper.
> > > >
> > > > Reviewer F4UP

---

### Author Response · Authors · 2022-11-14
**General response**

We thank the reviewers for their thoughtful and constructive review of our manuscript.
We notice that Reviewer rDHq, Reviewer MNku, and Reviewer F4UP have raised some concerns about the novelty of our work, and here we provide a unified response.

We will also respond to questions raised by each reviewer individually.

>Novelty

Our work does combine two existing methods: optimizable triggers and importance sample selection. However, it is definitely not a simple combination, but includes our thoughts on improving the poisoning efficiency. In addition to the existing selection and construction, we investigate the effect of an **unexplored** factor, randomness, on the poisoning efficiency of backdoor attacks and identify a good characteristic of this factor (individual consistency) that can be used to reduce the number of poisoned samples further. Efficient Trojan Injection (ETI) is presented based on the existing and our research for probing the capability limit that is currently achievable.

We believe that this impression may be partly due to the fact that in the original version we did not clarify the relationship between our work and other work, and that the final ETI results are not shown in Figure 2. We have made the appropriate changes in the revised version.

---

### Decision · Program_Chairs · 2023-01-20

**Decision:**

Reject

**Justification For Why Not Higher Score:**

Lack of enough novelty as discussed in part 1.

**Justification For Why Not Lower Score:**

N/A

**Metareview: Summary, Strengths And Weaknesses:**

The paper proposes Efficient Trojan Injection (ETI) to reduce the number of poisoned samples needed when backdooring an image classifier. The proposed technique combines two existing methods: optimizable triggers and importance sample selection. In addition, it investigates the effect of randomness, on the poisoning efficiency of backdoor attacks and exploits individual consistency to further reduce the number of poisoned samples. All the reviewers appreciate the importance of the problem and the good empirical results. However, in general they are concerned about the novelty of the paper. As optimizable triggers and importance sample selection has been explored before, the main novelty is exploiting individual consistency to further reduce the number of poisoned samples. However, the paper lacks enough discuss about this. The paper will benefit from a revision to add more discussion on this point.